# The value of ecosystem services in global marine kelp forests

Aaron M. Eger [1,2] ✉, Ezequiel M. Marzinelli[3,4,5], Rodrigo Beas-Luna [6], Caitlin O. Blain[7], Laura K. Blamey [8], Jarrett E. K. Byrnes [9], Paul E. Carnell [10], Chang Geun Choi [11], Margot Hessing-Lewis[12,13], Kwang Young Kim[14], Naoki H. Kumagai [15], Julio Lorda [16], Pippa Moore[17,18], Yohei Nakamura[19], Alejandro Pérez-Matus[20,21], Ondine Pontier [12], Dan Smale [22], Peter D. Steinberg[1,4,5] & Adriana Vergés [1,5]

While marine kelp forests have provided valuable ecosystem services for millennia, the global ecological and economic value of those services is largely unresolved. Kelp forests are diminishing in many regions worldwide, and efforts to manage these ecosystems are hindered without accurate estimates of the value of the services that kelp forests provide to human societies. Here, we present a global estimate of the ecological and economic potential of three key ecosystem services - fisheries production, nutrient cycling, and carbon removal provided by six major forest forming kelp genera (*Ecklonia, Laminaria, Lessonia, Macrocystis, Nereocystis*, and *Saccharina*). Each of these genera creates a potential value of between \$64,400 and \$147,100/hectare each year. Collectively, they generate between \$465 and \$562 billion/year worldwide, with an average of \$500 billion. These values are primarily driven by fisheries production (mean \$29,900, 904 Kg/Ha/year) and nitrogen removal (\$73,800, 657 Kg N/Ha/year), though kelp forests are also estimated to sequester 4.91 megatons of carbon from the atmosphere/year highlighting their potential as blue carbon systems for climate change mitigation. These findings highlight the ecological and economic value of kelp forests to society and will facilitate better informed marine management and conservation decisions.

"The number of living creatures of all Orders, whose existence intimately depends on the kelp is wonderful." – Charles Darwin 1845[1]

Vast underwater forests of kelp (defined here as brown macroalgae in the order Laminariales) along polar to subtropical coastlines have enormous value to peoples across multiple continents and eras. Archaeological excavations show how kelp forests facilitated southward travel for early peoples in the Americas some 20,000 years ago. During this migration, people relied on the food provided by kelp forests to survive[2]. Subsequently, ecological management of kelp forests has occurred since approximately 3000 BCE in the NE Pacific, with peoples regulating harvest and transplanting kelp to enhance growth and trap fish roe[3]. In the NW Pacific, kelp harvesting has played

an important role in Japanese, Korean, and Chinese economies since the 8th century, where it is eaten as food and supports a myriad of associated plants and animals, many of which are also harvested. In Europe, kelp has been used for many centuries to fertilize soil and increase crop yields, treat illnesses caused by iodine deficiency and, for many centuries, as the base in the production of soda ash[4]. In the 20th and 21st centuries kelp forests have become the main source of alginate (also known as algin from alginate-yielding seaweeds), a common food, medical and bioengineering additive[5]. Globally, kelp forests provide habitat for important fisheries of abalone, lobsters, reef fishes, and kelp itself[6]. Additionally, through their high productivity, kelp forests draw carbon from the atmosphere[7], release oxygen[8], and help

reduce marine nutrient pollution[9,10]. Long before Charles Darwin wrote his essay on the Patagonian kelp forests, these habitats provided essential services for human society that continue to this day.

The fact that kelp forests have cultural and socioeconomic importance is not disputed, but the magnitude and economic values of these ecosystems are poorly understood[11–13]. Relevant research on kelp forests to date has generally grouped kelp with other marine habitats as "coastal systems"[14], treated values from limited genera as representative of not just kelps but all macroalgae[15], or has not assigned a monetary value to the services provided[16]. This knowledge gap leads to an underappreciation of their contribution to nature and people. Since both the economic value of ecosystems and the recognition of their ecological and cultural importance are increasingly major considerations for conservation and natural resource management, the lack of value estimates for kelp ecosystems is a barrier to effective management and policy[17].

For example, societies are increasingly considering active kelp forest restoration and management strategies to combat regional declines in kelp forests[18,19]. However, restoration may not be pursued if the costs outweigh the perceived benefits[20]. Furthermore, while kelp forests are valued to some degree by ocean users[21,22], they are not perceived to be high-value ecosystems to the public[23,24], which can limit public support for kelp conservation and restoration[25,26]. Moreover, quantifying and valuing services provided by marine ecosystems is an important goal in the context of the UN Decade of Ocean Sciences, achieving the UN Sustainable Development Goals, growing the field of ocean accounting, and cost-benefit analyses[27–29].

Regional economic valuations of kelp forests which have incorporated various ecosystem services (e.g. harvest, fisheries, and tourism) have estimated regional kelp forests to be worth between $290 million (e.g. *Ecklonia* and *Laminaria* forests in South Africa)[30] and USD $540 million per year (e.g. *Lessonia* and *Macrocystis* forests in Central-Northern Chile)[12]. In Australia, Bennett et al. (2016)[23], valued the ~71,

000 km² of 'The Great Southern Reef', including the lobster and abalone fisheries largely supported by *Ecklonia* habitat, at ~ $7.3 billion USD per year; though this value included all marine habitats, not only kelp. However, the above estimates were not standardized per area and did not directly link fisheries production within kelp forests to their final value. Consequently, there are currently no quantitative estimates of the area-adjusted economic value of major kelp genera worldwide.

Here we analyse three ecologically and economically important ecosystem services provided by six dominant kelp genera across the world: *Ecklonia*, *Lessonia*, *Laminaria* (now *Saccharina* in some regions), *Macrocystis*, and *Nereocystis*. While the order Laminariales comprises 33 genera[31], many of which provide similar ecosystem functions, we focused on kelp genera with the most widespread abundance and distributions and those with the highest regional socio-ecological importance (e.g., dominant habitat formers with important associated fisheries)[10]. These genera are distributed across the Northern and Southern Pacific, Northern and Southern Atlantic, and parts of the Arctic and Southern Oceans, and encompass most of the global kelp distribution[10]. Within these genera we analysed three services that had market values reported: fisheries (i.e., secondary) production, carbon capture, and nutrient cycling, which past studies suggest comprise the most valuable market services provided by kelp forests[12,23,30].

We first detailed the extent of the biophysical services generated and then assigned open market values (the price an asset would fetch in a marketplace, converted to international dollars 2020) to each service (see methods). We then generated a range of biophysical and potential economic values provided by each genus across regions, per unit of area, per year (see methods). As a result, our work describes the capacity of global kelp forests' to supply ecosystem services[32]. This capacity is the potential economic value (herein value) as opposed to the realized value. Like previous authors who have adopted this approach for valuing natural systems[33–35], we focus on potential value because, though it generates a higher estimate of economic value than realized value[36], it creates an inventory of resources[37], highlights potential future value[38], can identify areas for protection and management[39], and generates awareness about the socioeconomic importance of an ecosystem[40]. Our analysis provides a global quantification of the core ecological services provided by kelp forests as well as a global economic assessment of those services.

## Results and Discussion

We included 1354 fish and-or invertebrate surveys at distinct times and locations across the six different kelp genera in eight different ocean regions (North-East Pacific, North-West Pacific, South-West Pacific, South-East Pacific, North-West Atlantic, North-East Atlantic, South Atlantic and Southern Ocean). We observed 1583 unique species within these surveys.

We also collected 74 measures of net primary production (NPP), 23 measures of carbon composition, 29 measures of nitrogen composition, and eight measures of phosphorus composition. These values were collected from the eight ocean regions, though sample size varied among regions (Supplementary Data 1).

We found that approximately 740 million people live within 50 km of a kelp forest.

### Fisheries production economic values

We found substantial variation in the fisheries values between the different genera and within genera by region (Fig. 1). Further, the economic value of the fisheries depended on the harvest rate. To obtain a range of values, we varied extractions rates between 20 and 70%[41,42], while using an average value of 38%[42]. The lowest mean annual fisheries production rate was 111 kg/Ha/year ($780/ Ha/year), for *Macrocystis* in the Southern Ocean. The highest mean fisheries biomass value was for *Laminaria/Saccharina* in the Northwest Atlantic (3187 kg/

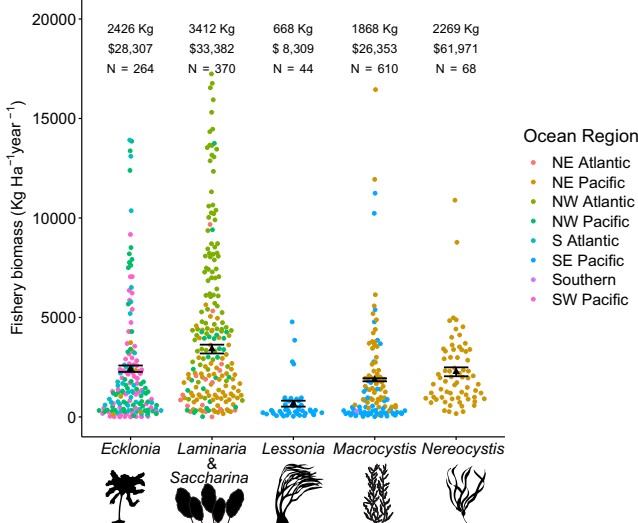

**Fig. 1 | Fisheries biomass and associated economic value provided by kelp forests.** Site (unique time and location) yearly total biomass and the economic value of the harvestable fisheries production per hectare per year. The values are represented for each kelp genus, colours represent the ocean region, the black triangle and number values represent the mean value for the genus, the error bars are the standard error. Note: the sample size represents the number of points used for calculation, though 70–90% of points for *Ecklonia*, *Laminaria* & *Saccharina* (previously classified as *Laminaria* in some regions), and *Macrocystis* have been randomly removed from the graph for better visualization. Image credit: Tim Carruthers, Integration and Application Network (ian.umces.edu/media-library) for the *Ecklonia*, *Laminaria*, *Lessonia*, *Macrocystis*, *Nereocystis* images.

Ha/year, $28,068/ha/year). Using our selected harvest ranges, 20 and 70%, the range of economic values were *Ecklonia* (15,040$–$52,645), *Laminaria/Saccharina* ($17,661–$61,810), *Lessonia* ($4420–$15,480), *Macrocystis* ($13,900–$48,610), and *Nereocystis* ($32,600–114,150) (Supplementary Data 2). Using a 38% harvesting rate, the economic values across ocean regions were: *Ecklonia*–922 kg ($28,307), *Laminaria/Saccharina*–1296 kg ($33,382), *Lessonia*–254 kg ($8309), *Macrocystis*–710 kg ($26,353), and *Nereocystis*–862 kg ($61,971) (Fig. 1, Supplementary Data 3). All values per hectare per year.

A relatively small number of genera comprised the bulk of the economic fisheries value at our sites. Indeed, only 50 genera from a total of 193 contributed more than an average of 10% of a site's economic fisheries production and 67 genera contributed more than 5%. On average, the most valuable genera were invertebrate species. These included lobsters (*Panulirus, Jasus, Hommarus*), abalone (*Haliotis*), false abalone "loco" (*Concholepas*), urchins (*Centrostephanus, Heliocidaris, Diadema, Strongylocentrotus, Loxechinus*), and crabs (*Necora, Cancer*) (Fig. 2). The most valuable reef and finfishes were pollack (*Pollachinus*), giant seabass (*Stereolepis*), South American morwongs (*Chirodactylus*), and lingcod (*Ophiodon*).

## Nutrient removal and carbon sequestration values

Bioremediation and carbon sequestration by kelp forests also provided substantial ecological benefits and economic value. The mean dollar value per hectare per year for the removal of carbon, nitrogen, and phosphorus is $36,109 for *Ecklonia*, $113,681 for *Laminaria/Saccharina*, $83,799 for *Lessonia*, $72,020 for *Macrocystis*, and $79,956 for *Nereocystis* (Fig. 3 split by ocean region). Of the three elements, nitrogen removal provided the highest economic value per hectare per year (mean = $73,831, 620 Kg), followed by phosphorus removal (mean = $4,075, 59 Kg), and lastly carbon capture (mean = $163, 720 Kg).

Carbon sequestration rates (see Methods) across genera and region varied by nearly an order of magnitude. Using a 10% sequestration rate estimate[15], the minimum regional average of carbon sequestration per m² per year was 31 g (*Ecklonia* in the South Atlantic) while the maximum was 214 g (*Macrocystis* in the Southern Ocean). Across genera, the average value (g/m²/year) per genus was 75 (*Ecklonia*), 109 (*Laminaria/Saccharina*), 151 (*Lessonia*), 101 (*Macrocystis*), 82 (*Nereocystis*). These values are dependent on the amount of NPP sequestered. If we assume a range of 1 and 20% of NPP sequestered[15], these values (g/m²/year) range from 7 to 150 (*Ecklonia*), 11 to 219 (*Laminaria/Saccharina*), 15 to 302 (*Lessonia*), 10 to 302 (*Macrocystis*), and 8 to 164 (*Nereocystis*). Considered globally over 30 years (to 2050), kelp forests would thus sequester between 14 and 292 megatons of carbon (Supplementary Data 4).

The removal rates for nitrogen and phosphorus varied by a factor of two to five. The average grams of nitrogen removed per m² per year were 41 (*Ecklonia*), 124 (*Laminaria/Saccharina*), 88 (*Lessonia*), 81 (*Macrocystis*), and 86 (*Nereocystis*), while the average grams of phosphorus removed per m² per year were 2 (*Ecklonia*), 13 (*Laminaria/Saccharina*), 16 (*Lessonia*), 5 (*Macrocystis*), and 12 (*Nereocystis*).

## Combined values

The average combined value per hectare per year of carbon storage, nutrient removal, and fisheries services ranged from $38,799 (*Macrocystis*, South-eastern Pacific) to $165,200 (*Laminaria*, North-western Atlantic), with an outlier value of $280,620 (*Laminaria/Saccharina*, North-western Pacific). Based on the kelp distributions in these areas (Supplementary Data 5), the regional value of kelp forests thus ranged from $0.66–157 billion per year (Fig. 4). Globally, these kelp forests produce an estimated average $500 billion per year with an Net Present Value of 7.44 trillion international dollars over the next 20 years (using a discount rate of 3%).

## Global Value of Kelp Forests

Global kelp forests generate considerable ecological and economic benefits across the world's oceans. Indeed, an estimated 740 million people live within 50 km of a kelp forest. These benefits vary according to the service being considered, the kelp genus, and the ocean region. In areas with available data, we found that the six genera annually generate between $1 and $157 billion per year regionally and totalled $500 billion globally. On a per-area basis, the values for each genus ranged from $64,700 and $147,300/hectare each year, and the average value across genera was $111,400. Previous work by Costanza et al. (2014), which considered nine ecosystem services and grouped algae with seagrass, valued those services at ~$36,000/hectare/year. As such, our estimate, which only considers kelp, and only three ecosystem services, is a >3-fold increase from the previous, best reported economic value of global kelp forests. These estimates are likely to increase when more services are considered.

We combined data on the spatial coverage of kelp forests (see methods) to provide a global economic estimate of the value of the selected kelp forests. While most regional estimates varied between $-1 and 130 billion per year, *Laminaria/Saccharina* in the North-western Atlantic was an exception to these values and was estimated to contribute $156 billion per year. The high value of *Laminaria* and *Saccharina* forests in the North Atlantic is attributable to its extensive distribution, covering 9500 km² in Eastern North America (Supplementary Data 5) and the large amounts of nitrogen that it removes, a service driven by its high primary productivity. Not all these services are converted to dollars (i.e., not all the fisheries production is removed and sold in a year and not all carbon capture or nutrient cycling is traded on markets), but these services have significant potential value to coastal economies. Past work suggests that non-market services like tourism and recreation can be the most economically important ecosystem service[43]. Adding these values to our estimate could thus substantially increase our estimates. Further, the regional estimates will increase as additional kelp genera are considered (e.g., *Alaria*, *Undaria*).

## Fisheries value

The potential fisheries value generated by kelp forests is substantial, with one hectare of underwater forest producing an average 2380 Kg/hectare/year, of which 904 kg is harvested when applying a 38% extraction rate. The average economic value of that 38% harvest is $29,851 per year, while a 20% harvest yields $15,771 a year and a 70% harvest yields $55,205 a year. Under these same scenarios, the global value of kelp forests shifts from $500 billion to $465 billion in the low harvest scenario and to $562 billion in the high harvest scenario.

These fisheries values only consider economically exploited species and do not consider the numerous kelp-associated organisms (1081 additional species in this study) that support other economically exploited components of the food web[6] or the species caught only in recreational fisheries. Of the economically important species, invertebrates such as lobster and abalone contributed the most fisheries value to kelp forests, often accounting for over 25% of the value of a site's fisheries. In fact, the abalone *Haliotis rufescens* contributed an average of 47% of a site's value for the genus *Nereocystis* (*N* = 56) and the mean economic fisheries value was highest for *Nereocystis*.

Kelp forests support biodiversity, with some species transiting through forests, others spending part of their life stage there, and others entirely obligate on the kelp forest[44]. Consequently, it is important to understand how much of the calculated fisheries value is directly attributable to kelp forests. Some of the most valuable genera in our study, e.g. *Panulirus*[45], *Jasus*[46], *Haliotis*[47], *Pollachius*[48], rely on kelp forests for habitat and food and declines in kelp populations have been linked to declines in these genera[49–51]. However, for some genera (e.g. *Homarus* and some sea urchins), loss of kelp forests has not always resulted in notable population declines[52–54].

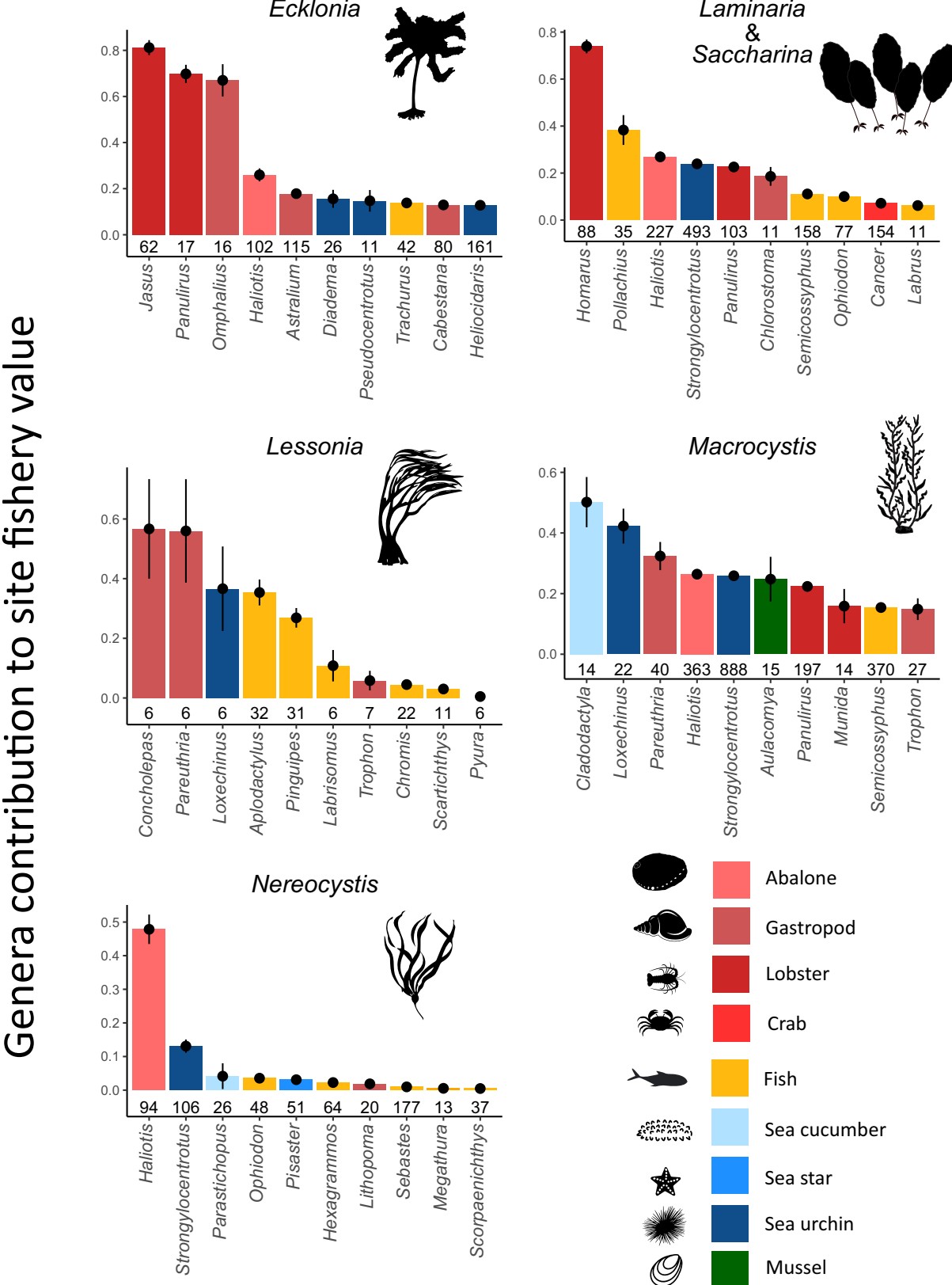

**Fig. 2 | Individual genera contributions to the economic value of study sites.** The mean proportion each genus contributed to a site's overall fisheries value per year, the lines represent plus and minus one standard error. The sample size is above the genera, n = number of surveys a genus appeared in, only genera that appeared in more than 10 surveys are represented (more than 5 for *Lessonia* due to fewer surveys). Image credit: Tim Carruthers, Integration and Application Network (ian.umces.edu/media-library) for the *Ecklonia, Laminaria, Lessonia, Macrocystis, Nereocystis*, abalone, snail, lobster, crab, fish, sea cucumber, sea star, sea urchin, and mussel images.

The exact contribution of kelp forest habitat to these fisheries services remains an important next step in understanding how kelp forests support food webs[55] and their related economies. A detailed review paper[16] revealed that kelp forests had a positive effect on fish abundance in 19 of the 24 studies reviewed, a positive effect on crustacean abundance in 4/4 studies, and a positive effect on gastropod abundance in 2/3 studies. Nevertheless, the exact amount of an organism's economic value directly attributable to kelp forests remains unresolved, we partially addressed this issue by assigning

genera into high, medium, low, and zero dependency categories and adjusting the economic value based on those classifications (see Methods, Supplementary Data 6). Future work could seek to further address this issue by using more detailed approaches such as stable isotope analysis.

For our economic evaluation, we aimed to value the sustainable harvestable fisheries biomass that is produced each year[56,57]. We chose this value over the total biomass produced to not promote the complete extraction of fisheries biomass and to enable the economic evaluation for consecutive years as opposed to a single year value (i.e., if all the biomass is removed in one year, there is no value left for the second year). Another alternative would be to report the realized value, i.e., the amount that is extracted, sold, and recorded by fisheries agencies.

While we chose to use the potential, sustainable value, records on fisheries landings provide an opportunity to examine how much of the service (secondary productivity) in kelp forests is being actively converted into a benefit (dollars). Such fisheries production estimates are available for some of the larger fisheries in areas with accurate records. For instance, the total value of wild fisheries in Australia are estimated to be worth $1032 million/year (2020)[58], whereas we estimated the fisheries production value of *Ecklonia* forests in Australia at $941 million/year (2020). Similarly, wild fisheries in California total ~$302 million/year[59] but we calculated that fisheries services for *Macrocystis* forests in the state are worth $1181 million/year. These potential values are roughly equal to four times the realized value of all fisheries in the respective regions. While valuable species like lobster and abalone are likely already fully exploited[41], there could therefore be new markets for other, currently less desirable species such as sea urchins[60]. The harvest rate will influence the realized economic value and what is sustainable will vary by species, region, and even year. Therefore, the harvest rates we used are only for illustrative purposes and should not be used to set fishing policy. While the realized economic fisheries value should always be less than the potential values, it is important to acknowledge that the unexploited biomass supports additional, currently unknown tourism values and continue to play an important part in the ecosystem[61,62].

Our work only quantifies those species that are directly consumed or sold by humans. It does not place a value on the species which play an important role in supporting the food web (e.g., forage fish), on

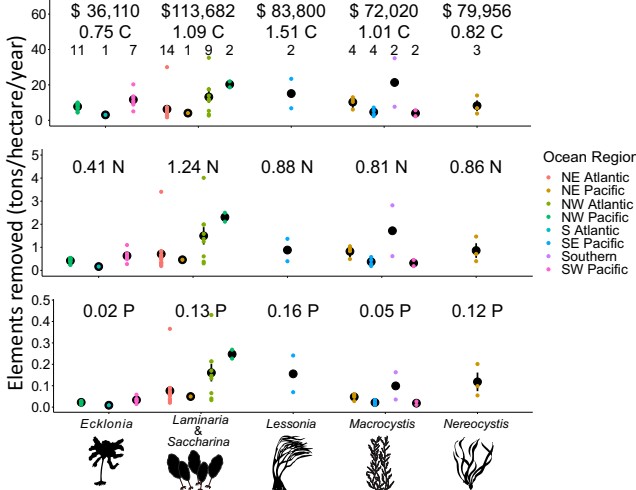

**Fig. 3 | Annual value of the carbon, nitrogen, and phosphorus removal services provided by kelp forests.** The mean yearly removal of carbon (C), nitrogen (N), and phosphorus (P) in tons per hectare per year. The black dots represent the mean value for the genus in that region, the error bars are the standard error. The currency is in thousands of international dollars for the year 2020 and is given as an average value for each genus. The top text dollar values are the combined economic value for the removal of all three elements. Sample sizes (unique location-time measurement) are presented above each point. Image credit: Tim Carruthers, Integration and Application Network (ian.umces.edu/media-library) for the *Ecklonia*, *Laminaria*, *Lessonia*, *Macrocystis*, *Nereocystis* images.

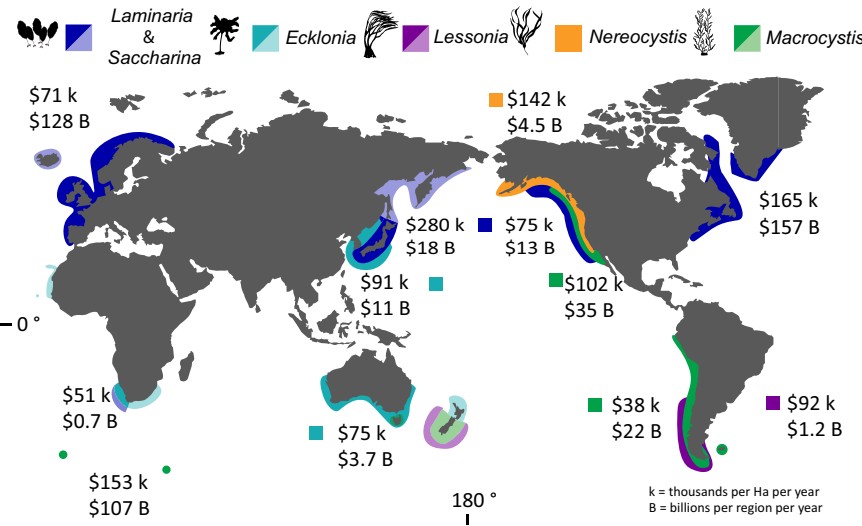

**Fig. 4 | Kelp forest distribution and associated economic value by region.** Map of kelp distribution, total economic value per m² per year (k), regional value (B). Lighter shade colours are for regions where distribution estimates were not available and therefore these values were not included in the regional value calculation.

Image credit: Tim Carruthers, Integration and Application Network (ian.umces.edu/media-library) for the *Ecklonia*, *Laminaria*, *Lessonia*, *Macrocystis*, *Nereocystis* images and map provided by FreeVectorMaps.com.

juvenile species that are not found within kelp forests as adults, or on the material value of the kelp itself. Obtaining accurate values for the associated fisheries services will be difficult but would increase the fisheries value of biodiversity in kelp forests once calculated. There are also a few remaining wild kelp harvest economies around the world, most notably in Chile[63], but also in Norway[49], Ireland[64], Mexico[65], and France[66] and these will add more value to kelp ecosystems. Indeed, a previous analysis of kelp forests in Chile found that wild harvest was 75% of the economic value in that region, while associated fisheries were only 15%[12]. We did not include the wild harvest value in this analysis because the industry is not consistently found for all genera in all regions but doing so would increase the regional value of kelp in the locations where those markets occur, namely in South America.

## Carbon sequestration

Using a sequestration rate of 10%[15], we found that the six kelp genera sequester between 31 and 214 g of carbon per $m^2$ per year. This rate of carbon sequestration is similar to other ecosystems. Terrestrial forest ecosystems report burial values of 54–120 $g/m^2/year$[67], seagrasses report ~83 $g/m^2/year$[68], mangroves ~174 $g/m^2/year$, and salt marsh ~150 $g/m^2/year$[69]. While the exact values are subject to change dependent on the year, location, and environmental conditions, these general comparisons suggest that kelp forests, which generally do not provide below ground carbon burial in the habitats where they grow, are comparable contributors to carbon sequestration in natural systems.

These values are, however, contingent on multiple mechanisms that influence carbon sequestration rate of kelp, such as consumption or decomposition after detachment[70], biotic interactions[71], prevailing winds, ocean currents and local topographies such as coastal marine canyons[72]. If the sequestration rate were reduced to 1%, the potential for carbon capture in kelp forests would be significantly reduced to averages between 7 and 15 $g/m^2/year$ depending on the kelp genus. Alternatively, if the sequestration rate were increased to 20%, kelp forests would be some of the best habitat for naturally capturing carbon, ranging, on average between 150 and 302 $g/m^2/year$. Further research addressing the fate and transport of kelp carbon to other habitats is needed to decrease the uncertainty associated with this range of potential sequestration values.

Putting these numbers into context shows that regional kelp forests sequester between 4,000 and 1.48 million tons of carbon per year. Because the area estimates we used are likely underestimates and did not account for deep water kelp, these values are conservative. Together, these six genera of kelp are estimated to sequester at least 4.91 megatons of carbon from the atmosphere per year. Taken over 30 years (e.g., 2050, a common climate goal), these kelp forests will sequester between 14 and 292 megatons of carbon (1–20% sequestration, Supplementary Data 7).

Filbee-Dexter and Wernberg[7] recorded a much higher potential (e.g. 1.3 megatons C/year for Australia compared to 4.91 megatons C/year globally in our study) for carbon sequestration. This mismatch is likely due in part to the differences in estimated areal distributions, as they assumed all rocky habitat as kelp forest. The other major study[15] estimated 173 megatons but accounted for yet unmapped deep sea kelp forests and considered all macroalgae in their estimates, resulting in values that are therefore not directly comparable to ours. Further, as the science of blue carbon in kelp forests continues to develop, new approaches and approximations will refine these results[73,74].

Interestingly, despite the high per $m^2$ carbon capture potential of kelp forests, the economic value of this ecosystem service in our study was low relative to other values. The mean economic value of carbon capture was only $163 per hectare per year even though we used the social cost of carbon (~$45/ton C[75]), a relatively high estimate that incorporates the social and environmental externalities of increased atmospheric $CO_2$ concentrations in our evaluation. Previous work

suggests that even the social cost of carbon underestimates the true value of carbon capture[76]. Nevertheless, even if the price of carbon were to increase ten-fold to $450/ton, the resulting economic value of carbon capture in kelp forests would remain relatively low at $1630/hectare/year. This outcome suggests people should use caution when promoting carbon capture as a purely economic incentive for restoring or protecting kelp forests or indeed other marine ecosystems.

## Nutrient removal

At an average value of $73,831/hectare/year, nitrogen removal from the water column was a more economically valuable service compared to drawdown of carbon or phosphorus. The high value is attributed to the proportionally high uptake of nitrogen compared to phosphorus, the high dollar value allocated to nitrogen removal, and the fact that nitrogen and phosphorus do not need to be transported to the deep sea to be effectively removed.

Placing an economic value on the nitrogen removed from the ocean requires some simplification. First, we obtained estimates of nutrient trading schemes from the Eastern United States, Southern Australia, and Europe. These schemes are based on the replacement cost of the service, that is, how much it would cost to build a water treatment plant to remove the same amount of nitrogen as the kelp. Our approach equates the ocean-based removal of these nutrients with these numbers. While there are inherent mechanistic differences between upstream and ocean-based removal, these equivalencies are necessary in the absence of market-based values for these processes[77]. Further, we present the amount of nitrogen that kelp takes up in a year and do not quantify the instantaneous removal rate. Therefore, our economic evaluation is based on the yearly amount of nitrogen removed by a kelp forest combined with the economic value of removing that amount of nitrogen before it enters the ocean. Altering either of these assumptions will alter the evaluation.

Nitrogen and phosphorus removal only results in direct benefits in areas with excessive nutrients, typically near rivers, agricultural regions, and urban areas[78] which also contain a kelp forest. Therefore, the realized value of nitrogen removal will be lower than the potential value described here. Conversely, this value may also increase as kelp forests in these zones would provide additional services and value by reoxygenating hypoxic zones that are often caused by nutrient pollution[79] and we have not included that. Further incorporating these complexities would increase the accuracy of our evaluations. Until that is possible, we suggest that the nutrient removal services only be considered in areas with elevated nutrients that still have kelp present. We include these services in our approximation of the value of kelp forests as they represent the potential value of kelp to a region, should those services be needed. Indeed, Froehlich et al. (2019)[80] found that 77 countries suitable for macroalgae growth have hypoxic, eutrophic, or acidic waters, signalling a high potential for the use of these services.

## Realized versus potential value

There are numerous ways to place an economic value on ecosystem services[81] and while estimating the potential value of ecosystems services is a common approach[14,82,83], other methods will result in different evaluations[84–86]. This fact is well demonstrated by the previous discussions on potential versus extracted fisheries values, and nutrient cycling and carbon capture when no one is paying for them (i.e., no credits are purchased or traded). While we made several adjustments to assess the direct economic contribution, few nutrient markets exist, carbon trading is not widely applied or validated for kelp forests, and not all fish biomass is extracted for market sale. Therefore, our values are higher than the direct current contribution of kelp forests to global markets (i.e., GDP). Rather the values presented in this study represent the biophysical services generated each year (tons of fish, and Kg of carbon, nitrogen, and phosphorus removed). We then obtain an economic value by attributing the current market price to those values.

We believe this approach highlights the global value of kelp forests, whether extracted or not, but acknowledge the results should not be used in decision making that is motivated only by realized economic outputs. Further work should continue to refine these values to account for realized value[38], marginal costs[87], and supply and demand[88].

### Drivers of variation

We found substantial variation in the ecosystem service values described in this study. This variation was found within and across genera and ocean region and was related to the services themselves, market pricing, and the spatial and temporal distribution of kelp forests. Market prices for the fish species will depend on the year, season, level of processing, distance to market, risk of spoilage, and other factors such as changes in regulation and governance[89–91]. Similarly, the price of carbon, nitrogen, and phosphorus will also change through time. As the market prices change, there will be corresponding changes to the estimated values presented here and these values are thus a snapshot.

Spatially, the North-eastern Pacific region had the most data points and therefore, the averages for *Macrocystis* and *Laminaria* are biased towards that region. To try to understand whether these imbalances might bias our estimates, we removed random portions of the data points in that region until the number of samples were comparable to the other genera. As a result, average fisheries value for *Macrocystis* dropped from ~$26,000/hectare/year to ~$20,000/hectare/year, reflecting the higher value of fisheries in the NE Pacific compared to other *Macrocystis* related fisheries in South America. Conversely, the fisheries value for *Laminaria* was little changed by this resampling.

Explaining the rest of the variation will be a key next step in predicting the value of a kelp forest. The services considered in our study are based on production, first of the kelp and second of its associated biodiversity. At the regional scale, we expect this production to be driven by nutrients, temperature, and photoperiod[92,93], while smaller scale differences maybe driven by depth, salinity, wave exposure, biotic pressures, and human stressors[24,94,95]. In an era of dynamic change due to impacts such as warming oceans, coastal development, it is crucial to evaluate the expected alterations to ecosystem services based on system-level drivers and pressures, addressing their consequences from both ecological and economic perspectives.

### Kelp distribution

The differences in kelp cover between regions were much higher than the differences between per area average production or economic value. Therefore, the regional and global value of kelp forests is largely dependent on the estimates of kelp distribution. Estimates of the distribution of kelp forests for this research are dependent on two factors. First, true changes in kelp forest cover, due to natural environmental factors (e.g., El Niño[96]) and anthropogenic factors (e.g., overharvesting[97], nutrient pollution[98], and human caused climate change[99]) may increase or (more likely) decrease the total contribution of kelp forests to human society. Kelp decline has already led to closures of important abalone fisheries[51,100] and our findings further quantify the losses that will be associated with further kelp forest decline. Secondly, our findings are also subject to measurement errors on kelp distribution. We used existing datasets to approximate the area covered by different kelp genera across global ocean regions (Supplementary Data 5). While some of these estimates are precise, such as the estimates for *Macrocystis* which relies on satellite remote sensing data[101], other estimates were based on multiple assumptions. For instance, *Ecklonia* coverage in Australia was approximated using the area covered by rocky reef and the average kelp percent cover from the Reef Life Survey dataset[102]. Notably, we could not find estimates of *Laminaria* coverage in Russia or Iceland, *Lessonia*, *Ecklonia*, or *Macrocystis* in New Zealand, and *Ecklonia* in the mid-Atlantic or parts of the Southern Atlantic (Western Southern Africa). As the areal distributions of forests are improved upon, our estimates of kelp's value to society will be refined.

### Kelp forests in the future

As kelp forests become increasingly threatened by multiple drivers[10] it is imperative that we understand their considerable economic contribution to human society. Our results represent the first global ecological and economic assessment of marketable kelp forest services. This evaluation is not intended to commodify kelp forests, which support immense arrays of life and many other ecosystem services, but rather we hope to draw attention to their importance and inform policy and management decisions where benefits of kelps might be an important factor. We found that kelp forests are on average over 3 times more valuable than previously acknowledged and expect these evaluations to increase as more market and non-market services are assessed. For instance, canopy forming kelps can provide coastal protection[103,104], decrease pH and facilitate other organisms[105], as well as provide cultural connections and support tourism and other recreational opportunities[21]. Though unassessed in our study, kelp farms may offer similar ecosystems services and could be compared to natural populations and potentially considered in future regional and global accounts. While climate mitigating services will continue to be an important field of investigation, we found that the greatest economic value of kelp forests was from fisheries production and uptake of nitrogen. As a result, we present these services as the best economic motivators for kelp conservation and restoration. These values situate the value of kelp forests among other marine ecosystems while providing a template for conducting similar analyses in unassessed ecosystems. As the field advances, it will be important to expand on these approximations and work to explain the variation documented in our baseline study.

## Methods

### Literature search and data collection

We conducted genera-specific literature searches to compile densities for fisheries species found in kelp forests, as well as net primary production (NPP, i.e., the amount of biomass accumulated in one year) and elemental composition (percent composition of carbon, nitrogen, and phosphorus) values for the six kelp genera (Supplementary Data 8). The first searches were conducted on Scopus Web of Science. We read selected papers in their entirety to ensure that they met our inclusion criteria, namely that they recorded the density of a commercially relevant species in kelp habitat, measured the average annual production or net primary production for the kelp species or reported a year averaged elemental composition of the same genera. If a paper met our criteria, we first assigned it to an oceanographic region, either North Eastern or Western Pacific, South Eastern or Western Pacific, the North Eastern or Western Atlantic, the Southern Atlantic, or the Southern Ocean. From each paper we recorded the mean density of fish or invertebrate associated with each genus, the mean net primary production, and the mean carbon, nitrogen, or phosphorus composition. Fisheries species were collected at any time during the year while NPP and percent elemental composition were collected as annual averages (Supplementary Data 1 and 9). Fish surveys were collected between the years of 1988–2020, came from 11 countries, ranging from 56° S to 71° N.

We collected additional biodiversity and NPP data from online repositories such as Reef Life Survey, Reef Check California, and the Hakai Institute. Because there were limited publicly available data in some regions, we sought out additional unpublished datasets directly from researchers in Australia, Chile, Korea, the North Atlantic, South Africa, and Japan. Datasets from Japan and the Eastern United States contain surveys for species once classified in the genus *Laminaria* but now in *Saccharina*, these data are included in our analysis as *Laminaria* and they are referenced together throughout this paper.

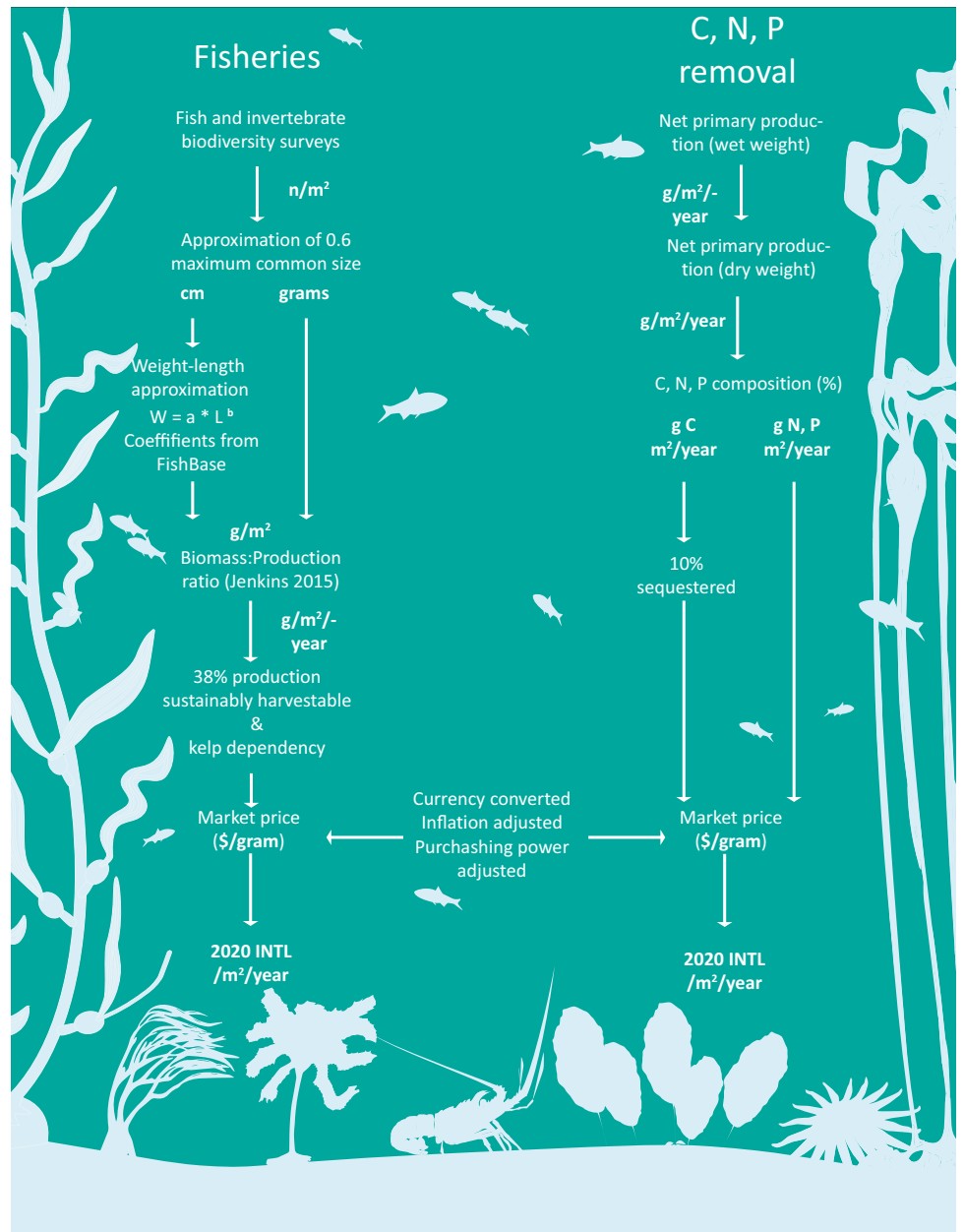

**Fig. 5 | Flow chart of steps for calculating the market value of different services.** Diagram of the data processing steps used to translate ecological values into economic ones. In weight-length approximation, W = weight, L = length, a & b = coefficients. Image credit: Tim Carruthers, Integration and Application Network (ian.umces.edu/media-library) for the *Ecklonia, Laminaria, Lessonia, Macrocystis, Nereocystis*, sea star, lobster, and fish images.

## Fisheries calculations

We estimated the secondary production of fish and invertebrates by using published values on species' length and weights (Supplementary Data 10) and a biomass to production relationship. Because most studies did not report a species' length or size, we first estimated a species' length at 60% of its recorded maximum length[106]. We opted to use the 60% estimate because not all species observed in each survey would have been the maximum size. We then calculated a species weight (grams) using established length-weight relationships[106]. If a species had no length or weight-length relationship values, we used values from species in the same genera or family. If there was no value available in the same genus or family, we searched for biomass estimates. After we obtained a species' biomass, we converted this value into production (grams per year) using a validated productivity-biomass relationship[107] (Fig. 5). To ensure a future harvest, not all fish production is harvested in one year. As a result, there is considerable

variation in reported sustainable harvest rates for fisheries[41,42]. Therefore, in our economic evaluation, we considered that a range from 20 to 70% of production is harvested each year while using an observed average value of 38%[42] as a base rate. The sustainable harvest level will vary by species, region, and time but these numbers cover the span of observed values.

We conducted repeated literature and internet searches to find species-specific market or wholesale values for the fish and invertebrates. We first checked FishBase to see if a species was used by humans[106] and considered all potential fisheries including commercial, recreational, and artisanal (Supplementary Data 11). If no fishery market value was reported on Fishbase, we conducted additional web searches to confirm this find. If after 50 Google and Google Scholar search results, we could not find a market value or indication of an active fishery, we considered that species to have no fisheries market value. If we found evidence of a fishery but could not find a value, we applied the

same taxonomic averaging approach as described for obtaining biomass. Species market values were recorded at differing levels of processing (e.g., dried versus alive) and some were sold for consumption while others were sold on the ornamental market. All values are recorded in the supplement (Supplementary Data 11). The fisheries values were then adjusted for purchasing power and converted into international dollars/Kg[14] and adjusted for inflation to the year 2020 (Fig. 5). If we found multiple values for a species, we took the average value.

Ultimately, we found market values for 502 of the observed 1583 species of fish and invertebrates with 395 from retail pricing, 76 from reports, 63 from peer reviewed literature, 18 from industry sources, 10 from news articles, 9 used genus averages, 9 from books, and 7 from webpages. The per kilo prices ranged from \$0.29 to \$324 and were collected from 32 countries. Because the amount of money invested before turning a profit varies by countries, we accounted for this "cost of capital" based on the country the fish was extracted from. These values ranged from 3–15% (Supplementary Data 12)[12,108,109]. Further, as the prices were obtained for products with different levels of processing (e.g., live versus filleted versus dried), we adjusted for the resources required for each processing type as well as the risk of that product spoiling and being worth nothing. The discount rate for a highly processed product or a likely to spoil product was 2.5%, therefore a maximum discount rate of 5% per price was applied (Supplementary Data 13). These values are partial corrections and were approximated due to the lack of available information. Given the uncertainty around these values, the discounts were approximated so that they were similar to the other discount values applied (e.g., cost of capital) and thus did not have an outsized influence on the results. Such cost adjustments may be improved upon in future analysis.

### Genus dependency on kelp forest habitat

Species maybe observed in a kelp forest but may not strongly or uniquely depend on kelp forest ecosystems for food, shelter, or other benefits. As such, their economic value may not be wholly attributable to a kelp forest ecosystem. We accounted for this fact by creating dependency classes for 187 genera of fish and invertebrates, adapted from[110,111]. We then used available information about a genera's habitat preferences and life history to qualitatively classify genera as either having a high, medium, low, or no dependency on kelp forests. We then corrected for the partial dependency of the medium and low classifications by attributing 2/3rd and 1/3rd of the total economic value respectively to kelp forests. If a species appeared in 5 or fewer surveys, we assigned the genus an economic value of zero as they were likely incidentally observed and not dependent on kelp forests. However, we included all observations in *Lessonia* habitat due to the limited number of data points available. These corrections are based on our expert opinion and are subject to change with further analysis (e.g., stable isotope, mixing models, experiments). All relevant data are presented in Supplementary Data 6.

Following these adjustments we obtained the annual fisheries value of kelp habitat by multiplying the species-specific productivity by the species-specific market value. Finally, we assessed the range of site values per ocean region.

### Carbon sequestration and nutrient removal

We used the average elemental composition of each genus as reported in the literature to convert region specific NPP into the average amount of carbon, nitrogen, and phosphorus absorbed from the water each year (Supplementary Data 1). Because not all fixed carbon is permanently removed from the water column, we used a tentative estimate that 10% of kelp NPP is exported to the deep sea and effectively removed from the system[15,112]. While this estimated percentage is the best available, it remains to be validated. This value represents the amount of carbon that is removed from the atmosphere over a prolonged period (>100 years). It is the value that is most relevant to

carbon trading schemes and for evaluating mitigation of carbon dioxide emissions associated with anthropogenic climate change. Because the exact sequestration value is undetermined, we also ran a sensitivity analysis to account for alternative sequestration values (1–20% sequestration, Supplementary Data 7).

We collected market prices for the social cost of carbon and averaged nutrient trading schemes from around the world (Supplementary Data 14). The social cost of carbon reflects the environmental and social costs (e.g., crop failure, damage from sea level rise) that are caused by emitting an additional ton of carbon into the atmosphere. It is typically higher than market schemes (e.g. cap and trade or taxes) but is increasingly being pressed for as a price that reflects the consequences of climate change[75,76]. The value of nitrogen and phosphorus removal were calculated as the mean of the available prices for removal of a kilogram of that element (Supplementary Data 14). The prices themselves are calculated by determining how much a society would have to invest in infrastructure to prevent a kilogram of nitrogen or phosphorus from entering the ocean and are reflective of nutrient trading schemes in the USA, Australia, and Europe[113–115].

We then multiplied the yearly amount of carbon, nitrogen, and phosphorus removed by the averaged dollar costs to obtain the value of these ecosystem services (Fig. 5). As with the fisheries values, we assessed site values by ocean region.

All dollar values in our analysis are presented in international dollars for the year 2020 and have been adjusted using the purchasing power exchange rate[116], unless stated otherwise.

### Spatial distribution of kelp

We compiled existing estimates of the spatial coverage of kelp forests in each region as well as calculated new approximations for regions where specific survey data was available (Supplementary Data 5). The data collection methods included in this compilation ranged from remote sensing[101], government reports from aerial images[117], to combinations of percent cover[118] and suitable kelp habitat (e.g., rocky reef and depth)[119].

We combined the estimated spatial niche of kelp forests[10] with coastal human population data from 2020[120] to estimate how many people lived within 50 km of a kelp forest.

### Net present value

The net present value was calculated using a 3% discount rate[121,122] and represents the current present value of 20 years of services provided by 1 hectare of kelp forest (i.e., potential economic value from 2021 to 2041)[123].

### Reporting summary

Further information on research design is available in the Nature Portfolio Reporting Summary linked to this article.

## Data availability

The data generated in this study have been deposited in the Open Science Framework database under accession code osf.io/ykqc3/. The data are also presented in the Supplementary Dataset 1 file.

## Code availability

All the scripts required to run the analysis on this project are available for download from an Opens Science Framework depository located at osf.io/ykqc3/. All analysis was done in the R programming language (V4.0.0) with RStudio (V1.4.17.17).

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

## Acknowledgements

The authors would like to acknowledge Jess Nguyen, Fumi Hayashi, Michelle Kenney, Anna Roniotis, Sabina Hamilton, Sara Abbey, Gurinder Singh, Tiarne Harris, Rosemary Steinberg, Camila Gonzalez-Alonso de Linaje, and Tahsin Khan, all of whom helped us collate and validate the fisheries cost data and the species weight-length functions. AE, AV, DS, and PM would also like to acknowledge our continued collaboration with the GEAK network and our valued discussions on how to value ecosystem services in kelp forests. In particular, we thank Johanna Zimmerhackel, Cristina Pineiro-Corbeira, Kjell-Magnus Norderhaug, Karen Filbee-Dexter, and Thomas Wernberg for their thinking and work on fisheries dependencies in kelp forest ecosystems. The work was also supported by a Scientia PhD scholarship from the University of New South Wales to AE and was partly supported by an Australia Research Council Discovery grant to AV (DP190100058). O. Pontier and M. Hessing-Lewis were supported by the Tula Foundation and the Hakai Institute. For the Gulf of Maine KEEN ONE data, we'd like to thank Doug Rasher, Marissa MacMahon, Jon Grabowski, Austin Humphries, Jenn Dijkstra, Caitlin Cleaver, Madison Maier, and their respective dive teams.

## Author contributions

A.E., P.S., E.M. and A.V. conceptualized the research. A.E. designed the study, coordinated co-authors, collated the data, ran the analysis, and wrote the first draft. R.B.L., L.B., C.B., J.B., P.C., C.G.C., M.H.L., K.Y.K., N.K., J.L., P.M., Y.N., A.P.M., O.P., D.S. all contributed unpublished data to the analysis. All authors reviewed the manuscript, edited the contents, approved it for submission, and participated in the peer review process.

## Competing interests

We declare that the authors have no competing interests as defined by Nature Portfolio, or other interests that might be perceived to influence the results and/or discussion reported in this paper.

## Additional information

[1]School of Biological, Earth, and Environmental Sciences, University of New South Wales, Sydney, NSW 2033, Australia. [2]Kelp Forest Alliance, Sydney, NSW 2034, Australia. [3]The University of Sydney, School of Life and Environmental Sciences, Sydney, NSW, Australia. [4]Singapore Centre for Environmental Life Sciences Engineering, Nanyang Technological University, Singapore, Singapore. [5]Sydney Institute of Marine Science, Mosman, NSW, Australia. [6]Universidad Autónoma de Baja California, Facultad de Ciencias Marinas, Ensenada, BC, Mexico. [7]Leigh Marine Laboratory, Institute of Marine Science, University of Auckland, Auckland, New Zealand. [8]Commonwealth Scientific and Industrial Research Organization, Environment, Brisbane, QLD 4072, Australia. [9]Department of Biology, University of Massachusetts Boston, Boston, MA 20125, USA. [10]School of Life and Environmental Sciences, Deakin University, Queenscliff, VIC 3225, Australia. [11]Department of Ecological Engineering, Pukyong National University, Busan, South Korea. [12]Hakai Institute, Quadra Island, Canada. [13]Institute of the Oceans and Fisheries, University of British Columbia. 2202 Main Mall, Vancouver, BC V6T 1Z4, Canada. [14]Department of Oceanography, College of Natural Sciences, Chonnam National University, Gwangju 61186, Korea. [15]Center for Climate Change Adaptation, National Institute for Environmental Studies, 16-2 Onogawa, Tsukuba, Ibaraki 305-8506, Japan. [16]Universidad Autónoma de Baja California, Facultad de Ciencias, Ensenada, BC, Mexico & The Tijuana River National Estuarine Research Reserve, Imperial Beach, CA, USA. [17]School of Life Sciences, Aberystwyth University, Aberystwyth SY23 3DA, UK. [18]Dove Marine Laboratory, School of Natural and Environmental Sciences, Newcastle University, Newcastle-upon-Tyne NE1 7RU, UK. [19]Graduate School of Integrated Arts and Sciences, Kochi University, 200 Monobe, Nankoku, Kochi 783-8502, Japan. [20]Subtidal Ecology Laboratory (Subelab), Estación Costera de Investigaciones Marinas (ECIM), Departamento de Ecología, Facultad de Ciencias Biológicas, Pontificia Universidad Católica de Chile, Casilla 114-D Santiago, Chile. [21]Millennium Nucleus for the Ecology and Conservation of Temperate Mesophotic Reef Ecosystem (NUTME), Las Cruces, Valparaiso, Chile. [22]Marine Biological Association of the United Kingdom, Citadel Hill, Plymouth PL1 2PB, UK. ✉e-mail: aaron.eger@unsw.edu.au

