## [Peer Review File · Nature Communications]

The value of ecosystem services in global marine kelp forestsReviewer #1 (Remarks to the Author):

Summary:

This study provides the first global economic valuation of ecosystem services provided by kelp forests, focusing on three main ecosystem services (fisheries production, nutrient cycling, and carbon removal) and four major forest forming kelp genera (Macrocystis, Nereocystis, Ecklonia, and Laminaria). Compiling a database of site level area-specific estimates for these services, the authors then analyze patterns of global variation in ecosystem service value across regions and genera using publicly available literature, datasets, and source.

Importantly, global estimates indicate that kelps systems are producing five times the economic value found in previous studies. The findings also suggest that economic values from kelps are primarily driven by fisheries production and nitrate removal services rather than blue carbon storage, mainly because of the lucrative nature of fisheries, the higher cost of nitrogen remediation, and the comparatively low estimated 'social cost' of carbon. Still, global estimates suggest that kelps are capturing roughly 2.7 megatons of carbon per year and that they are likely still providing significant non-monetary value as important blue carbon reservoir.

In my opinion, this study addresses an important gap in the marine ecosystem service valuation literature. To date, many ecosystem service valuation studies have tended to focus on regional scales and/or combine estimates for kelps with other macroalgae or "coastal habitat" categories. Studies related to blue carbon have also largely overlooked kelps, focusing instead on biogenic habitats that predominantly store carbon in their root systems and sediments (e.g., seagrasses, marshes, and mangroves). This paper would make a nice contribution to the literature. I think results could be of interest to both researchers and policy-making working in fields of marine resource management, conservation, and restoration. I just have a few comments and concerns that need to be addressed prior to publication.

Major Comments:

First, I think the authors have done a tremendous amount of work assembling datasets on the supply and value of fisheries production, nutrient uptake, and carbon sequestration from four kelp genera. However, I think the study may be providing a geographically biased estimate of service values by excluding other important and widely distributed genera (e.g, Saccharina, Alaria and Lessonia; Teagle et al. 2017). Can the authors provide more explicit justification for choosing these four genera over others (beyond data availability)? It might also likely strengthen the paper to acknowledge how regional estimates of service values might change with the inclusion of these other genera.

On a related note, it looks like the fisheries production data is pretty heavily biased towards the Pacific. As such, the analysis likely does a much better job of characterizing variation within and across genera in the Pacific regions compared to others. I am thinking specifically about the fisheries production from kelp in the northeastern and western Atlantic. Some of the highest value fisheries in the US, Canada, and Europe come from species that occupy kelp habitat (e.g., groundfish and American lobster). But figure 1 appears to show pretty sparse, low value estimates for these regions. Have the authors considered incorporating estimates from government reports or reaching out to government agencies with fishery independent monitoring programs within kelp-dominated systems in U.S., Canada, and/or other countries bordering the Atlantic? Or at least accounting for the unbalanced nature of the dataset in your analysis?

Lastly, I have concerns about data consistency, particularly related to the kelp area estimates and the market value estimates for individual fish and invertebrate species. While some of the sources seem to be reliable coming from published studies or government reports, other sources seem more difficult to verify and may be less accurate (e.g., personal communications and internet forum posts). I suggest reconsidering some of these sources and potentially only using sites estimates coming

from consistent and verifiable sources. Specifically related to fish value, the authors could consider using the global ex-vessel fish price database instead of mark value (Sumaila et al. 2007).

Minor Comments

Line 36: Is there a more direct citation you can use here?

Fig. 1. This is minor, but the legend labels need to be fixed. Right now, they currently say "ne_atlantic" etc. for northeast Atlantic.

Also, in fig 1, its very difficult to see the data points from the Atlantic regions because of the color scheme and density of Pacific points. I suggest using a different color scheme and potentially jittering the datapoints or using separate boxplots (with samples sizes) for different regions in order to make the plot easier to read .

Line 146: Is there a reason that value estimates for Ecklonia from west Africa were excluded from this map?

Lines 167: How did you determine the area occupied by specific genera in each of the regions? Please clarify this in the methods section. Do these areas include kelp habitats that are potentially occupied by excluded genera? Or do these estimates only reflect the known distribution of the four genera?

Line 228: The Fillbee-Dexter & Wernberg 2020 citation is missing from the list of references.

Line 235: It's quite an interesting finding that the monetary value from fisheries and nutrient uptake in kelps drastically outweighs the value of carbon storage. However, it's not totally clear what factors are getting accounted for in the "social cost of carbon" estimate. So, it's a little difficult for the reader to assess how much this actually underestimates the "environmental damage that is being caused by carbon". I would suggest providing more detail about the "social cost of carbon" measurement in the methods section. You might also consider adding carbon value estimates based on the cost of carbon removal using other methods as a comparison.

Lines 332: Its unclear how you dealt with fish production estimates from different gear types. It strikes me that the gear types targeting fishes are likely different from those targeting lobster and crabs. Species also commonly have different catchability rate between gear types. This could influence both your fish production estimates (fig 1) and the proportion of site value estimate for different species (fig. 2). It would strengthen the paper to add more detail about this in the methods and/or discussion section.

References:

- Teagle, Harry, et al. "The role of kelp species as biogenic habitat formers in coastal marine ecosystems." *Journal of Experimental Marine Biology and Ecology* 492 (2017): 81-98.
- Sumaila, U. Rashid, et al. "A global ex-vessel fish price database: construction and applications." *Journal of Bioeconomics* 9.1 (2007): 39-51.

Reviewer #2 (Remarks to the Author):

The authors have collated a substantial global data set to examine the value of key ecosystem services provided by kelp forests. The aims of this paper are important, the contribution is well articulated, and the results are likely to be of broad interest for the field. However, I do have some issues with the approach that should be clarified. I also find the presentation of the findings is quite spun up to support a narrative of the incredible value of these ecosystems, and several important uncertainties and

background assumptions are skimmed over. As it is currently presented this is misleading and could lead to problematic use.

Overall the work would be stronger if it presented the biophysical measures of the 3 ecosystem services, and then applied a possible market value to these. Currently the abstract and much of the main text only focus on the market values and economic values. For example, fisheries are presented as proportion of site value and per ha and per y value. But data on how many commercial fish use these habitats are not clearly shown. Same is true for carbon and nitrogen. An overall estimate of total C storage or N filtration per ha would be highly useful, but these services are only presented in terms of USD per year. There are numerous assumptions and steps required to translate biophysical measures to economic values, and these alone are often rarely sufficient for describing the entire value of an ecosystem service (needing to be combined with say human use). Skipping over the presentation of the biophysical measures means that much of the useful data is buried in supplement. This is highlighted in the discussion - to properly look at how much of the service (secondary productivity) is converted to benefit (dollars) – and here it would be useful to know the catch effort for these species, compared to the total densities.

The authors take an approach of calculated the potential value of services provided by kelp forests, as opposed to the realized value (e.g., the actual flow of services to humans). This is somewhat articulated in the discussion, but should be stated clearly in the abstract. There is an important distinction between these two types benefits (IPBES 2020, Hein et al., 2016; Jones et al., 2016; Villamagna et al., 2013). These two different valuations cannot for example, be both used in comparisons across other ecosystems. Calculating a 'total potential fisheries production value' for a service using existing market values for that species weight is not as straightforward as the authors seems to imply. Supply and demand factors should be taken into account. This creates some problems with their approach of taking the per area densities of all commercially fished species within a kelp forest, multiplying them by the total kelp forest area and then by the average market value for that species, using current fisheries in that region (often very small scale fisheries). This, not surprisingly, produces large values for a single species that are sometimes more than the entire country or state value of all commercial fisheries. Some of these species are not even fished in the areas they are taking the density data from.

For example, if there are a billion kg of urchins in the sea, but market is only for half a million kg, it is problematic to say that fishery is potentially worth a billion kg in value. This follows a scenario where we are catching ALL the urchins. It also could be interpreted to mean than we are under harvesting species in the kelp forests, which is potentially dangerous. Ideally a measure of profit associated with the fishery should be used.

Second, there needs to be a better link between co-occurrence of commercial species and data showing they rely on the kelp forest. Are these species using the kelp forest in a certain way? Or are they only associated with the rocky reef structure? See for example the lobster comment below.

Minor comments are listed below:

Abstract. Line 12: while I agree in principle that they have potential, I wonder if this conclusion is beyond the scope of the data presented. Considering these as classic blue carbon systems requires more than sequestration estimates. It requires restoration tools or the ability to influence carbon sequestration at scale. I am not sure this is yet the case for kelp forests.

Line 20. Polar coasts? Are there Laminariales in Antarctica?

Line 36. 'towering' is quite emotive language. Unless Darwin used this term I would stick to more scientific descriptors.

Line 38. Reference for this line? Some of these services are quite disputed.

Line 49. To many ocean users (or too many kelp users?). Do you need a reference for this?

Line 54/63. What about the work that came out of MERCES project on ecosystem services in Norway?

Line 69. Of the global kelp distribution

Line 73. What economic tools and frameworks were used to assign market values to each service? Are their established practices for this from other systems.

Line 91-96. How was fisheries production linked to the presence of the kelp forest?

Fig 2. In some areas of western Canada, the abalone fishery actually wanted for kelp forests to remain in a degraded barrens state, because it was better for the fishery. This is a detail, but is there a way to account for these negative relationships between commercial species and kelp?

Line 122. Where does the 1-20% range come from?

Line 165. This statement gives me concern. In eastern Canada there is strong evidence that the lobster are not very dependent on the kelp forests, but are linked to the locations that the kelp are in. Much of the fishery is coming from Bay of Fundy, and often offshore. In fact, when kelps declined in eastern Canada due to warming temperature, there was no impact on the lobster, the stock actually increased in abundance. Evidence to date suggests that the one of the main limiting factors for the lobster fishery are deeper cobble nursery habitats.

A. Serdyska and S. Coffen-Smout. Mapping Inshore Lobster Landings and Fishing Effort on a Maritimes Region Statistical Grid (2012–2014). Oceans and Coastal Management Division. Ecosystem Management Branch. Fisheries and Oceans Canada. Bedford Institute of Oceanography

Scheibling RE 1994. Interactions among lobsters, sea urchins and kelp in Nova Scotia, Canada. In: Echinoderms Through Time, Proc. 8th International Echinoderms Conference, Dijon, France. David B, Guille A, Feral J-P, Roux M (eds) A.A. Balkema, Rotterdam, pp. 865-870

Line 183. It is important here to highlight this is the potential fisheries value, not the actual value. To get an economic value landing data would be more appropriate.

Line 376. Were the regional costs of nitrogen removal used, or a global average?

Line 385. Is there a reference for this approach?

Appendices: the Scheibling et al. 1999 reference for sea urchin densities is dated. There are currently little to no sea urchins in this region due to disease, and the sea urchin fisheries closed in the last decade from lack of supply. Disease as a control of sea urchin populations in Nova Scotian kelp beds. CJ Feehan, RE Scheibling. Marine Ecology Progress Series 500, 149-158. The values used are for NFL, a different region, which is largely in a barrens state (Work by P. Gagnon, Memorial University).

REVIEWER COMMENTS

Reviewer #1 (Remarks to the Author):

*We thank the reviewer for their constructive feedback and think their suggestions have worked to improve the quality of the work. We have endeavoured to add substantial new data and include an additional genus with *Lessonia*, as well as regions with *Laminaria* in the NW Atlantic, *Macrocystis* in the Southern Ocean, and *Laminaria* in the NE Pacific- Response 1. Further, we have reassessed the pricing data used and removed the prices from personal communications and forums, though these steps did not noticeably change our results – Response 2 and 3. We then incorporated all the new numbers into the manuscript text and figures.*

We provide our individual responses to the reviewer below and have denoted the sections as follows

Red: Reviewer's comment

Blue: Our response

Black: In text changes made

Summary:

This study provides the first global economic valuation of ecosystem services provided by kelp forests, focusing on three main ecosystem services (fisheries production, nutrient cycling, and carbon removal) and four major forest forming kelp genera (*Macrocystis*, *Nereocystis*, *Ecklonia*, and *Laminaria*). Compiling a database of site level area-specific estimates for these services, the authors then analyze patterns of global variation in ecosystem service value across regions and genera using publicly available literature, datasets, and source.

Importantly, global estimates indicate that kelps systems are producing five times the economic value found in previous studies. The findings also suggest that economic values from kelps are primarily driven by fisheries production and nitrate removal services rather than blue carbon storage, mainly because of the lucrative nature of fisheries, the higher cost of nitrogen remediation, and the comparatively low estimated 'social cost' of carbon. Still, global estimates suggest that kelps are capturing roughly 2.7 megatons of carbon per year and that they are likely still providing significant non-monetary value as important blue carbon reservoir.

In my opinion, this study addresses an important gap in the marine ecosystem service valuation literature. To date, many ecosystem service valuation studies have tended to focus on regional scales and/or combine estimates for kelps with other macroalgae or "coastal habitat" categories. Studies related to blue carbon have also largely overlooked

kelps, focusing instead on biogenic habitats that predominantly store carbon in their root systems and sediments (e.g., seagrasses, marshes, and mangroves). This paper would make a nice contribution to the literature. I think results could be of interest to both researchers and policy-making working in fields of marine resource management, conservation, and restoration. I just have a few comments and concerns that need to be addressed prior to publication.

Major Comments:

First, I think the authors have done a tremendous amount of work assembling datasets on the supply and value of fisheries production, nutrient uptake, and carbon sequestration from four kelp genera. However, I think the study may be providing a geographically biased estimate of service values by excluding other important and widely distributed genera (e.g., *Saccharina*, *Alaria* and *Lessonia*; Teagle et al. 2017). Can the authors provide more explicit justification for choosing these four genera over others (beyond data availability)? It might also likely strengthen the paper to acknowledge how regional estimates of service values might change with the inclusion of these other genera.

1) Given the diversity of genera within the order Laminariales, we needed to select a subset to make the work feasible. We selected the 4 original genera (Ecklonia, Laminaria, Nereocystis, and Macrocystis) based on their local abundance and widespread distribution, which covers nearly all regions of the world where kelp is found. For instance, Ecklonia is the most widespread kelp in the Southern hemisphere, Laminaria is the most widespread in the Northern, Macrocystis is the most widespread surface canopy former globally, Nereocystis is a significant canopy former in North America, and Lessonia is the dominant genus in South America (Wernberg et al. 2018). However, we understand the desire to be comprehensive and following the reviewer's recommendation, we have added a new genus (Lessonia) and have substantially expanded our dataset with 89 new site observations for Saccharina species (previously classified as Laminaria in the Atlantic) as well as 45 new fisheries sites for the Lessonia in South America. Each of these additions are paired with the corresponding primary production and distribution data. We believe these additions substantially increase the breadth and robustness of our work.

L69-77

Here we summarize ecologically and economically important ecosystem services provided by six dominant kelp genera across the world: *Ecklonia*, *Lessonia*, *Laminaria* (now *Saccharina* in some regions), *Macrocystis*, and *Nereocystis*. While the order Laminariales comprises 13 genera, many of which provide similar ecosystem functions, we focussed on kelp genera with the most widespread abundance and distributions and those with the highest regional socio-ecological importance (e.g., dominant habitat formers, important associated fisheries). These

genera are distributed across the Northern and Southern Pacific, Northern and Southern Atlantic, and parts of the Arctic and Southern Oceans, and encompass most of the global kelp distribution⁹.

Further to the reviewer's point, we have acknowledged in the Discussion that estimates will change as more genera are considered

L199-200

Further the regional estimates will increase as additional kelp genera are considered (e.g., *Alaria*, *Undaria*).

On a related note, it looks like the fisheries production data is pretty heavily biased towards the Pacific. As such, the analysis likely does a much better job of characterizing variation within and across genera in the Pacific regions compared to others. I am thinking specifically about the fisheries production from kelp in the northeastern and western Atlantic. Some of the highest value fisheries in the US, Canada, and Europe come from species that occupy kelp habitat (e.g., groundfish and American lobster). But figure 1 appears to show pretty sparse, low value estimates for these regions. Have the authors considered incorporating estimates from government reports or reaching out to government agencies with fishery independent monitoring programs within kelp-dominated systems in U.S., Canada, and/or other countries bordering the Atlantic? Or at least accounting for the unbalanced nature of the dataset in your analysis?

*2) The reviewer raises an important point, and it was one that we strived to address as we collated data prior to submission. However, because the Pacific Ocean has many more long term monitoring programs, we were able to compile significantly more information from those regions. We have addressed this issue in two ways. First we added data from grey literature, reports and papers published in Spanish and unpublished surveys to fill in the gaps in the Atlantic and Pacific. Second, to standardize the results based on the unbalanced nature of our revised data set, we have randomly removed 50-90% of the *Macrocystis* and *Laminaria* points from the North East Pacific region (W North America) to evaluate the impact on the outcome. The impact for *Laminaria* was negligible though the average value of fisheries for *Macrocystis* does drop from 65k to 51k when these points are removed. This drop reflects the fact that there are higher value fisheries in the North East Pacific than in the South East region. We have now noted this fact in our discussion*

L341-350

Some of this variation relates to variation in fisheries species present in an area, market prices, NPP, and imbalances in the dataset. The North-eastern Pacific region had the most data points and therefore, the averages for *Macrocystis* and *Laminaria*

are biased towards that region. To try to understand whether these imbalances might bias our estimates, we removed random portions of the data points in that region until the number of samples were comparable to the other genera. As a result, average fisheries value for *Macrocystis* dropped from ~\$66,000/hectare/year to ~\$51,000/hectare/year, reflecting the higher value of fisheries in the NE Pacific compared to other *Macrocystis* related fisheries in South America. Conversely, the fisheries value for *Laminaria* was little changed by this sampling.

Lastly, I have concerns about data consistency, particularly related to the kelp area estimates and the market value estimates for individual fish and invertebrate species. While some of the sources seem to be reliable coming from published studies or government reports, other sources seem more difficult to verify and may be less accurate (e.g., personal communications and internet forum posts). I suggest reconsidering some of these sources and potentially only using sites estimates coming from consistent and verifiable sources. Specifically related to fish value, the authors could consider using the global ex-vessel fish price database instead of mark value (Sumaila et al. 2007).

3) Thank you for the suggestion, we have now studied the Sumaila dataset in more detail. We unfortunately do not think it is fit for our approach because it aggregates fisheries values at large species groups (e.g. demersal, clam, and oyster) whereas our dataset is structured on species-specific surveys. We acknowledge that prices from the published literature are more desirable but our work would be severely limited if we only included such estimates. Further, we think the compilation of our price dataset provides a useful resource for indexing the available information on the retail or wholesale value of these species.

To address this comment and better report all sources of information, we have added a column to our dataset which categorizes the data source for each price estimate and includes a description of this in text (below). This tabulation shows very few prices (N = 18/579) come from websites or forums and the majority come from retail shops, which while unpublished are linked and dated (year) for verification. We have now removed these 18 data points from our analysis and this step had a negligible impact on our results (<100\$/hectare/year changes)

L451-453

Ultimately, we found market values for 561 species of fish and invertebrates with 366 from retail pricing, 85 from reports, 46 from peer reviewed literature, 23 from industry sources, 17 from news articles, 9 used genus averages, 8 from books, and 7 from webpages.

Minor Comments

Line 36: Is there a more direct citation you can use here?

4) We have added references which are explicit to the specified ecosystem services.

L39

6. Filbee-Dexter, K. & Wernberg, T. Substantial blue carbon in overlooked Australian kelp forests. *Sci. Rep.* **10**, 1–6 (2020).
7. Hatcher, B. G., Chapman, A. R. O. & Mann, K. H. An annual carbon budget for the kelp *Laminaria longicuris*. *Mar. Biol.* **44**, 85–96 (1977).
8. Kim, J. K., Kraemer, G. P. & Yarish, C. Use of sugar kelp aquaculture in Long Island Sound and the Bronx River Estuary for nutrient extraction. *Mar. Ecol. Prog. Ser.* **531**, 155–166 (2015).

Fig. 1. This is minor, but the legend labels need to be fixed. Right now, they currently say “ne_atlantic” etc. for northeast Atlantic.

5) We agree, this is preferable and has been adjusted based on the reviewer’s suggestion (see below).

Also, in fig 1, its very difficult to see the data points from the Atlantic regions because of the color scheme and density of Pacific points. I suggest using a different color scheme and potentially jittering the datapoints or using separate boxplots (with samples sizes) for different regions in order to make the plot easier to read .

6) We have randomly reduced the number of points displayed on the graph by 50 % to make it more readable. We don’t think using separate plots would be as effective as the genera are present in different ocean regions.

Fig. 1: Site (unique time and location) biomass and economic value of fisheries production per hectare per year. The values are represented for each kelp genus, colours represent the ocean region, the black triangle and number values represent the mean value for the genus, the error bars are the standard error. Note, the sample size represents the number of points used for calculation, though 70-90% of points for *Ecklonia*, *Laminaria*, and *Macrocystis* have been randomly removed from the graph for better visualization.

Line 146: Is there a reason that value estimates for *Ecklonia* from west Africa were excluded from this map?

7) Thank you, we have added transparent shading to the map to reflect that *Ecklonia* does occur in these regions but we did not collect data for these regions. We did not originally

include estimations from Ecklonia in West Africa as we did not find any biodiversity data from those locations nor did we find any distribution data.

Figure 4: Map of kelp distribution, total economic value per m² per year (k), regional value (B). Lighter shade colours are for regions where distribution estimates were not available and therefore these values were not included in the regional value calculation.

Lines 167: How did you determine the area occupied by specific genera in each of the regions? Please clarify this in the methods section. Do these areas include kelp habitats that are potentially occupied by excluded genera? Or do these estimates only reflect the known distribution of the four genera?

8) We have worked to better clarify our methods for calculating the area occupied. We used the mapped area of kelp distribution across the seafloor, not the potential distribution based on their range. These estimates are based on explicit values from other mapping surveys and our own calculations from existing datasets. We have clarified this in text in the methods as follows.

L491-495-494

Spatial distribution of kelp

We compiled existing estimates of the spatial coverage of kelp forests in each region as well as calculated new approximations for regions where specific survey data was

available (Appendix 3). The data collection methods included in this compilation ranged from remote sensing⁹⁴, government reports from aerial images¹⁰⁵, to combinations of percent cover¹⁰⁶ and suitable kelp habitat (e.g., rocky reef and depth)¹⁰⁷.

We have also added a section in the discussion describing how changes in time and estimations may alter our results.

L360-379

Kelp distribution

The differences in kelp cover between regions were much higher than the differences between per area average production or economic value. Therefore, the regional and global value of kelp forests is largely dependent on the estimates of kelp distribution. Estimates of the distribution of kelp forests for this research are dependent on two factors. First, true changes in kelp forest cover, due to natural environmental factors (e.g., El Niño⁸⁹) and anthropogenic factors (e.g., overharvesting⁹⁰, nutrient pollution⁹¹, and human caused climate change⁹²) may increase or (more likely) decrease the total contribution of kelp forests to human society. Kelp decline has already led to closures of important abalone fisheries^{18,93} and our findings further quantify the losses that will be associated with further kelp forest decline. Secondly, our findings are also subject to measurement errors on kelp distribution. We used existing datasets to approximate the area covered by different kelp genera across global ocean regions (Appendix 3). While some of these estimates are precise, such as the estimates for *Macrocystis* which relies on satellite remote sensing data⁹⁴, other estimates were based on multiple assumptions. For instance, *Ecklonia* coverage in Australia was approximated using the area covered by rocky reef and the average kelp percent cover from the Reef Life Survey data set⁹⁵. Notably, we could not find estimates of *Laminaria* coverage in Russia or Iceland, *Lessonia*, *Ecklonia*, or *Macrocystis* in New Zealand, and *Ecklonia* in the mid-Atlantic (Western Africa). As the areal distributions of forests are improved upon, our estimates of kelp's value to society will be refined.

Line 228: The Fillbee-Dexter & Wernberg 2020 citation is missing from the list of references.

9) Thank you, we have corrected this omission

Line 235: It's quite an interesting finding that the monetary value from fisheries and nutrient uptake in kelps drastically outweighs the value of carbon storage. However, it's not totally clear what factors are getting accounted for in the "social cost of carbon" estimate. So, it's a little difficult for the reader to assess how much this actually underestimates the "environmental damage that is being caused by carbon". I would suggest providing more detail about the "social cost of carbon" measurement in the methods section. You might also consider adding carbon value estimates based on the cost of carbon removal using other methods as a comparison.

10) We have worked to further explain the background and factors related to the social cost of carbon in the methods as suggested.

L472-476

The social cost of carbon reflects the environmental and social costs (e.g., crop failure, damage from sea level rise) that are caused by emitting an additional ton of carbon into the atmosphere. It is typically higher than market schemes (e.g. cap and trade or taxes) but is increasingly being pressed for as a price that reflects the consequences of climate change^{72,73}.

11) We hope this answers the reviewer's questions about the price estimate. Trading scheme values such as these ones are the most commonly used values for carbon evaluation, so we still think this is the most appropriate. We have run the requested analysis using the cost of carbon removal as a comparison and found that it does not significantly change our results (carbon capture technology is estimated at 12 – 350 USD/ton) so we believe it is not needed as an additional result in our paper.

We outline this implication in

L296-299

Nevertheless, even if the price of carbon were to increase ten-fold to \$500/ton, the resulting economic value of carbon capture in kelp forests would remain relatively low at \$1,630/ hectare/year. This outcome suggests caution in using carbon capture as a purely economic incentive for restoring or protecting kelp forests or indeed other marine ecosystems.

Lines 332: Its unclear how you dealt with fish production estimates from different gear types. It strikes me that the gear types targeting fishes are likely different from those targeting lobster and crabs. Species also commonly have different catchability rate between gear types. This could influence both your fish production estimates (fig 1) and the proportion of site value estimate for different species (fig. 2). It would strengthen the paper to add more detail about this in the methods and/or discussion section.

12) Our fish production values were not dependent on fisheries catch data or gear types. Rather we converted biodiversity surveys, known lengths and weights with a biomass to production formula (Jenkins 2015) to calculate secondary production. For the economic valuation, we used the potential price of the ecosystem services per year. These estimates are therefore not based on the realised value (or benefit) as fisheries catch per year, but rather represent the value if they were caught by fishers. Our cost calculations also use the

retail or wholesale value of the different species and thus reflect the differing costs associated with harvesting different species. We have expanded on these points in the text below

L82-90

We then generated a range of biophysical and potential economic values provided by each genus across regions, per unit of area, per year (see methods, figure 5). As a result, our work describes global kelp forests' capacity to supply ecosystem services³⁰, and their potential economic value (herein value) as opposed to the realized value (often contrasted as natural capital versus human derived capital³¹). Like previous authors who have adopted this approach for valuing natural systems³²⁻³⁴, we focus on potential value because, though it generates a high estimate of economic value³⁵, it creates an inventory of resources³⁶, highlights potential future value³⁷, can identify areas for protection and management³⁸, and generates awareness about the socio-economic importance of an ecosystem³⁹.

L222-224

We present the potential value of fisheries biomass that is produced each year^{50,51}, as opposed to the realized value, i.e. the amount that is extracted, sold, and recorded by fisheries agencies.

L440-441

We conducted repeated literature and internet searches to find species specific market or wholesale values for the fish and invertebrates.

L428-439

We estimated the secondary production of fish and invertebrates by using published values on species' length and weights (Appendix 6) and biomass to production relationship. Because most studies did not report a species' length or size, we first estimated a species' length at 50% of its recorded maximum length⁹⁸. We opted to use the 50% reduction because not all species observed in each survey would have been the maximum size. We then calculated a species weight (grams) using established length-weight relationships⁹⁸. If a species had no length or weight-length relationship values, we first looked for the values from species in the same genera or family. If there was no value available in the same genus or family, we searched for published biomass estimates. If biomass estimates were unavailable, we took averages from the species' class, order, or phyla. After we obtained a species' biomass, we converted this value into production (grams per year) using a validated productivity-biomass relationship⁹⁹ (Figure 5).

13) We also created a new figure to better highlight how these values were calculated (Fig 5).

Figure 5: Flow chart of methodological steps for calculating the market value of different services.

References:

- Teagle, Harry, et al. "The role of kelp species as biogenic habitat formers in coastal marine ecosystems." *Journal of Experimental Marine Biology and Ecology* 492 (2017): 81-98.
- Sumaila, U. Rashid, et al. "A global ex-vessel fish price database: construction and applications." *Journal of Bioeconomics* 9.1 (2007): 39-51.

Reviewer #2 (Remarks to the Author):

A) We thank the reviewer for their constructive feedback and think their suggestions have worked to improve the quality of the work. We have endeavoured to expand on and highlight the difference between potential and realized values throughout the manuscript – Response D. We have also presented the biophysical results along with the economic results through the manuscripts – Response C. Further, we acknowledge the difficulties attributing value to species which may only spend part of their life in a kelp forest and discuss these implications – Response E. We have addressed all of the reviewer’s other comments as well.

We provide our individual responses to the reviewer below and have denoted the sections as follows

Red: Reviewer’s comment

Blue: Our response

Black: In text changes made

The authors have collated a substantial global data set to examine the value of key ecosystem services provided by kelp forests. The aims of this paper are important, the contribution is well articulated, and the results are likely to be of broad interest for the field. However, I do have some issues with the approach that should be clarified.

We thank the reviewer for their constructive feedback and think their suggestions (listed below) have worked to improve the quality of the work.

I also find the presentation of the findings is quite spun up to support a narrative of the incredible value of these ecosystems, and several important uncertainties and background assumptions are skimmed over. As it is currently presented this is misleading and could lead to problematic use.

B) We appreciated the reviewer raising this issue and have worked to mollify this problem throughout the manuscript. Mainly, we have highlighted the differences between realized

and potential value for fisheries, carbon, and nutrient capture throughout the document. These changes are noted in the individual responses below.

Overall the work would be stronger if it presented the biophysical measures of the 3 ecosystem services, and then applied a possible market value to these. Currently the abstract and much of the main text only focus on the market values and economic values. For example, fisheries are presented as proportion of site value and per ha and per y value. But data on how many commercial fish use these habitats are not clearly shown.

Same is true for carbon and nitrogen. An overall estimate of total C storage or N filtration per ha would be highly useful, but these services are only presented in terms of USD per year. There are numerous assumptions and steps required to translate biophysical measures to economic values, and these alone are often rarely sufficient for describing the entire value of an ecosystem service (needing to be combined with say human use). Skipping over the presentation of the biophysical measures means that much of the useful data is buried in supplement. This is highlighted in the discussion - to properly look at how much of the service (secondary productivity) is converted to benefit (dollars) – and here it would be useful to know the catch effort for these species, compared to the total densities.

C) We agree that being more explicit about the biophysical data adds further transparency to the results as well as avoid solely evaluating these ecosystems in economic terms. To achieve this, we have remade figures 1 and 3 to include the biophysical values and changed the text in a number of places, examples of which are given below

Figure 1

Fig. 1: Site (unique time and location) biomass and economic value of fisheries production per hectare per year. The values are represented for each kelp genus, colours represent the ocean region, the black triangle and number values represent the mean value for the genus, the error bars are the standard error. Note, the sample size represents the number of points used for calculation, though 70-90% of points for *Ecklonia*, *Laminaria*, and *Macrocystis* have been randomly removed from the graph for better visualization.

Figure 3

Figure 3: The mean yearly removal of carbon (C), nitrogen (N), and phosphorus (P) in tons per hectare per year. The black dots represent the mean value for the genus in that region, the error bars are the standard error. The currency is in thousands of USD for the year 2020 and is given as an average value for each genus. The top text dollar values are the combined economic value for the removal of all three elements. Sample sizes (unique location-time measurement) are presented above each point.

L69-71

Here we summarize ecologically and economically important ecosystem services provided by six dominant kelp genera across the world: *Ecklonia*, *Lessonia*, *Laminaria* (now *Saccharina* in some regions), *Macrocystis*, and *Nereocystis*.

80-84

We first detailed the extent of the biophysical services generated and then assigned open market values (the price an asset would fetch in a marketplace) to each service (see methods, figure 5). We then generated a range of biophysical and potential economic values provided by each genus across regions, per unit of area, per year (see methods, figure 5).

L112-119

The lowest mean annual fisheries production rate was 252 kg/Ha/year (\$5,851/ Ha/year), for *Macrocystis* in the Southern Ocean, the highest mean production value was 6,762 kg/Ha/year (\$95,590/Ha/year) for *Laminaria/Saccharina* in the Northwest Atlantic, while the highest economic fishery value was for *Nereocystis* forests in the Northeast Pacific (\$159,167/Ha/year). The mean values across ocean regions were as follows: *Ecklonia* – 1,732 kg (\$72,381), *Laminaria/Saccharina* - 2,583 kg (\$85,694), *Lessonia* - 579 kg (\$29,601), *Macrocystis* – 1,325 kg (\$65,709), and *Nereocystis* - 1,838 kg (\$159,167), all values per hectare per year (Figure 1, Appendix 2).

L152-156

The removal rates for nitrogen and phosphorus varied over half an order of magnitude. The average grams of nitrogen removed per m² per year were 41 (*Ecklonia*), 124 (*Laminaria/Saccharina*), 88 (*Lessonia*), 81 (*Macrocystis*), and 86 (*Nereocystis*), while the average grams of phosphorus removed per m² per year were 2 (*Ecklonia*), 13 (*Laminaria/Saccharina*), 16 (*Lessonia*), 5 (*Macrocystis*), and 12 (*Nereocystis*).

L202-204

The potential value generated by fisheries species occupying kelp forests is substantial, with one hectare of underwater forest producing an average 1,752 kg/hectare/year, with an average value of \$76,009 per year.

The authors take an approach of calculated the potential value of services provided by kelp forests, as opposed to the realized value (e.g., the actual flow of services to humans). This is somewhat articulated in the discussion, but should be stated clearly in the abstract. There is an important distinction between these two types benefits (IPBES 2020, Hein et al., 2016; Jones et al., 2016; Villamagna et al., 2013). These two different valuations cannot for example, be both used in comparisons across other ecosystems. Calculating a 'total potential fisheries production value' for a service using existing market values for that species weight is not as straightforward as the authors seems to imply. Supply and demand factors should be taken into account.

This creates some problems with their approach of taking the per area densities of all commercially fished species within a kelp forest, multiplying them by the total kelp forest area and then by the average market value for that species, using current fisheries in that region (often very small scale fisheries). This, not surprisingly, produces large values for a single species that are sometimes more than the entire country or state value of all commercial fisheries. Some of these species are not even fished in the areas they are taking the density data from.

For example, if there are a billion kg of urchins in the sea, but market is only for half a million kg, it is problematic to say that fishery is potentially worth a billion kg in value. This follows a scenario where we are catching ALL the urchins. It also could be interpreted to mean that we are under harvesting species in the kelp forests, which is potentially dangerous. Ideally a measure of profit associated with the fishery should be used.

D) We agree with the reviewer's point to better articulate the distinction between the potential value and the realized value of these ecosystems. We have noted that these values are an over estimate, that the realized values should be expected to be lower, that this finding should not be used to justify over or total harvesting, and that they can be built upon with the addition of information such as supply and demand. We however, still present the potential values as past research has done in order to create an inventory of potential value, highlight the importance of these ecosystems, and generate awareness. We believe this is an important first step in the process of evaluating kelp forests.

We have made these changes throughout the manuscript, from the abstract to the discussion. Notably, we have added an entirely new section detailing the limitations of using the potential value.

Abstract

We present the first global estimate of the ecological and economic potential of three key ecosystem services - fisheries production, nutrient cycling, and carbon removal provided by six major forest forming kelp genera (*Ecklonia*, *Laminaria*, *Lessonia*, *Macrocystis*, *Nereocystis*, and *Saccharina*). Each of these genera creates a potential value of between \$112,900 and \$248,500/ ha each year.

We have provided justification for why we use the potential value in the introduction and highlighted that it will provide an overestimate

As a result, our work describes global kelp forests' capacity to supply ecosystem services³⁰, and their potential economic value (herein value) as opposed to the realized value (often contrasted as natural capital versus human derived capital³¹). Like previous authors who have adopted this approach for valuing natural systems³²⁻³⁴, we focus on potential value because, though it generates a high estimate of economic value³⁵, it creates an inventory of resources³⁶, highlights potential future value³⁷, can identify areas for protection and management³⁸, and generates awareness about the socio-economic importance of an ecosystem³⁹.

We have noted the importance of raising awareness for the value of these ecosystems

L42-43

That kelp forests have cultural and socioeconomic importance is not disputed, but the economic values of these ecosystems are poorly understood¹⁰⁻¹².

We also now acknowledge the potential limitations of our approach

L196-199

Not all these services are converted to dollars (i.e., not all the fisheries production is removed and sold in a year and not all carbon capture or nutrient cycling is traded on markets), but these services provide significant value to coastal economies.

We highlight that the realized values will be lower than the potential values

L239-242

While the realized economic fisheries value will be significantly less than our values, it is important to acknowledge that the unexploited biomass supports additional, currently unresolved tourism values and continue to play an important part in the ecosystem^{56,57}.

L323-337

Realized versus potential value

There are numerous ways to place an economic value on ecosystem services⁷⁷ and while the potential value of ecosystems services is a common approach^{13,78,79}, other methods will result in different evaluations⁸⁰⁻⁸². This fact is well demonstrated by the previous discussions on potential versus extracted fisheries values, and nutrient

cycling and carbon capture when no one is paying for them (i.e., no credits are purchased or traded). Because few nutrient markets exist, carbon trading is not widely applied or validated for kelp forests, and not all fish biomass is extracted for market sale, our values are higher than the direct contribution of kelp forests to global markets (i.e., GDP). Rather the values presented in this study represent the biophysical services generated each year (tons of fish and Kg of carbon, nitrogen, and phosphorus removed) and attributes the current market price to those values. We believe this approach highlights the value of kelp forests, whether extracted or not, but acknowledge the results are less accurate for use in decision making that is motivated by direct economic inputs. Further work should continue to refine these values to account for realized value³⁷, marginal costs⁸³, and supply and demand⁸⁴.

Further to this point, we note that we used a conservative approach by using half the maximum length or biomass of a species in our calculations. Therein, it is likely that there is more extractable biomass than we have estimated.

Second, there needs to be a better link between co-occurrence of commercial species and data showing they rely on the kelp forest. Are these species using the kelp forest in a certain way? Or are they only associated with the rocky reef structure? See for example the lobster comment below.

E) This is another important point raised by the reviewer. We had considered a by-species evaluation of the reliance and habitat uses of the different species involved in the study but determined this was beyond the scope of our work and would potentially create artificial cut offs or categorizations (e.g. what determines obligate use?).

Rather we have now worked to provide examples of kelp loss preceding biodiversity loss and added text to reflect that these species reside in kelp as adults and may have differing degrees of association to the habitat specifically.

L212-221

Kelp forests support an array of biodiversity, with some species transiting through forests, others spending part of their life stage there and others entirely obligate on the kelp forest. Consequently, it is important to understand how much of the calculated fisheries value is directly attributable to kelp forests. Some of the most valuable genera in our study, e.g. *Panulirus*⁴⁰, *Jasus*⁴¹, *Haliotis*⁴², *Pollachius*⁴³, rely on kelp forests for habitat and food and declines in kelp populations have been linked to declines in these genera^{18,44,45}. However, for some genera (e.g. *Homarus* and some sea urchins), loss of kelp forests has not always resulted in notable population declines^{46,47,48}. Partitioning the exact contribution of kelp forest habitat to these

fisheries services remains an important next step in understanding how kelp forests support food webs⁴⁹.

Minor comments are listed below:

Abstract. Line 12: while I agree in principle that they have potential, I wonder if this conclusion is beyond the scope of the data presented. Considering these as classic blue carbon systems requires more than sequestration estimates. It requires restoration tools or the ability to influence carbon sequestration at scale. I am not sure this is yet the case for kelp forests.

F) The reviewer has raised a good point and we have corrected the text to clarify their potential as blue carbon ecosystems.

L12-15

These values are primarily driven by fisheries production (1,752 Kg/Ha/year) and nitrogen removal (657 Kg N/Ha/year), but kelp forests also are estimated to sequester 4.9 megatons of carbon from the atmosphere/year and therefore have the potential to be considered blue carbon systems valuable for climate change mitigation.

Line 20. Polar coasts? Are there Laminariales in Antarctica?

G) Not on the mainland but on many sub-Antarctic Islands, which we think meets the criteria for this term but we can remove if still desired

Line 36. 'towering' is quite emotive language. Unless Darwin used this term I would stick to more scientific descriptors.

H) We agree with the reviewers assessment and have removed this descriptor

Line 38. Reference for this line? Some of these services are quite disputed.

I) We have added a reference and reworded the line to more clearly illustrate our point

L42-43

That kelp forests have cultural and socioeconomic value is not disputed, but the economic and biophysical values of these ecosystems are poorly understood¹⁰⁻¹²

Thurstan, R.H., Brittain, Z., Jones, D.S., Cameron, E., Dearnaley, J., Bellgrove, A., 2018. Aboriginal uses of seaweeds in temperate Australia: an archival assessment. *J. Appl. Phycol.* 30, 1821–1832.

Smale, D. A., Burrows, M. T., Moore, P., O'Connor, N. & Hawkins, S. J. Threats and knowledge gaps for ecosystem services provided by kelp forests: a northeast Atlantic perspective. *Ecol. Evol.* 3, 4016–4038 (2013).

Vasquez, J. A. *et al.* Economic valuation of kelp forests in northern Chile: values of goods and services of the ecosystem. *J. Appl. Phycol.* 26, 1081–1088 (2014).

Line 49. To many ocean users (or too many kelp users?). Do you need a reference for this?

J) We have now included two references on willingness to pay for kelp forest habitat

Hynes, S., Chen, W., Vondolia, K., Armstrong, C. & O'Connor, E. Valuing the ecosystem service benefits from kelp forest restoration: A choice experiment from Norway. *Ecol. Econ.* 179, 106833 (2021).

Grover, I. M., Tocock, M. S., Tinch, D. R. & MacDonald, D. H. Investigating public preferences for the management of native and invasive species in the context of kelp restoration. *Mar. Policy* 132, 104680 (2021).

Line 54/63. What about the work that came out of MERCES project on ecosystem services in Norway?

K) We believe the reviewer is referring to the following citation

Hynes, S., Chen, W., Vondolia, K., Armstrong, C., & O'Connor, E. (2021). Valuing the ecosystem service benefits from kelp forest restoration: A choice experiment from Norway. *Ecological Economics*, 179, 106833.

Hynes et al. used a willingness to pay design to evaluate people's perceptions towards kelp forests which does not translate well to the regional and country level evaluations of kelp forests discussed in lines 54 and 63. We have however incorporated this reference in the text above

L53-54

Furthermore, while kelp forests are valued to some degree by ocean users^{20,21} ...

Line 69. Of the global kelp distribution

L) We have made this correction

Line 73. What economic tools and frameworks were used to assign market values to each service? Are their established practices for this from other systems.

M) We have reworded the sentence to better reflect the methodology. We used the expressed market values to put a dollar value on the services and did not calculate a market value ourselves.

L77-79

Within these genera we analysed three services that had market values reported: fisheries (i.e., secondary) production, carbon capture, and nutrient cycling...

Line 91-96. How was fisheries production linked to the presence of the kelp forest?

N) We associated the production of fisheries species by their presence in a kelp forest. While we acknowledge that this approach does not partition how different species use and rely on kelp forests, we do think it reflects that kelp habitats exist in a mosaic and even if species do use other habitat as well, kelps play an important role in their life cycle. We have revised the below paragraphs to make this link clearer.

L403-411

We conducted genera-specific literature searches to compile densities for fisheries species found in kelp forests, as well as net primary production (NPP, i.e., the amount of biomass accumulated in one year) and elemental composition (percent composition of carbon, nitrogen, and phosphorus) values for the six kelp genera (Appendix 4). The first searches were conducted on Scopus Web of Science. We read selected papers in

their entirety to ensure that they met our inclusion criteria, namely that they recorded the density of a commercially relevant species in kelp habitat (target genus percent cover > 10%), measured the average annual production or net primary production for the kelp species or reported a year averaged elemental composition of the same genera.

L212-221

Kelp forests support an array of biodiversity, with some species transiting through forests, others spending part of their life stage there and others entirely obligate on the kelp forest. Consequently, it is important to understand how much of the calculated fisheries value is directly attributable to kelp forests. Some of the most valuable genera in our study, e.g. *Panulirus*⁴⁰, *Jasus*⁴¹, *Haliotis*⁴², *Pollachius*⁴³, rely on kelp forests for habitat and food and declines in kelp populations have been linked to declines in these genera^{18,44,45}. However, for some genera (e.g. *Homarus* and some sea urchins), loss of kelp forests has not always resulted in notable population declines^{46,47,48}. Partitioning the exact contribution of kelp forest habitat to these fisheries services remains an important next step in understanding how kelp forests support food webs⁴⁹.

Fig 2. In some areas of western Canada, the abalone fishery actually wanted for kelp forests to remain in a degraded barrens state, because it was better for the fishery. This is a detail, but is there a way to account for these negative relationships between commercial species and kelp?

O) This an interesting point, thank you for raising it. We think that our results are still relevant in this context though. The numbers reflect the value of abalone living in kelp forests and while in some instances, there may be more abalone living in barren habitat, the value estimated here is representative of the values in a kelp forest, and the regional estimations are based on area of kelp forests extent and do not include barren areas.

We have further expanded on this in the discussion as previously noted in response D

L217-221

However, for some genera (e.g. *Homarus* and some sea urchins), loss of kelp forests has not always resulted in notable population declines^{46,47,48}. Partitioning the exact contribution of kelp forest habitat to these fisheries services remains an important next step in understanding how kelp forests support food webs⁴⁹.

Line 122. Where does the 1-20% range come from?

P) We have added a reference for an expected range of these values (Krause-Jensen and Duarte 2016).

Line 165. This statement gives me concern. In eastern Canada there is strong evidence that the lobster are not very dependent on the kelp forests, but are linked to the locations that the kelp are in. Much of the fishery is coming from Bay of Fundy, and often offshore. In fact, when kelps declined in eastern Canada due to warming temperature, there was no impact on the lobster, the stock actually increased in abundance. Evidence to date suggests that the one of the main limiting factors for the lobster fishery are deeper cobble nursery habitats. A. Serdynska and S. Coffen-Smout. Mapping Inshore Lobster Landings and Fishing Effort on a Maritimes Region Statistical Grid (2012–2014). Oceans and Coastal Management Division. Ecosystem Management Branch. Fisheries and Oceans Canada. Bedford Institute of Oceanography

Scheibling RE 1994. Interactions among lobsters, sea urchins and kelp in Nova Scotia, Canada. In: Echinoderms Through Time, Proc. 8th International Echinoderms Conference, Dijon, France. David B, Guille A, Feral J-P, Roux M (eds) A.A. Balkema, Rotterdam, pp. 865-870

*Q) We have altered this section to remove this discussion point as we agree the link is not as strong and we think the section is much improved by focusing on the large spatial extent of *Laminaria* spp. in Europe and North America instead. We have also noted the variable habitat dependence of *Homarus* as above*

L194-196

The high value of *Laminaria* and *Saccharina* forests in the North Atlantic can be attributed to its high distribution, covering 9,500 km² in Eastern North America (Appendix 3) and its high primary production, which equates to high rates of valuable nitrogen cycling.

L217-221

However, for some genera (e.g. *Homarus* and some sea urchins), loss of kelp forests has not always resulted in notable population declines^{46,47,48}. Partitioning the exact contribution of kelp forest habitat to these fisheries services remains an important next step in understanding how kelp forests support food webs⁴⁹.

Line 183. It is important here to highlight this is the potential fisheries value, not the actual value. To get an economic value landing data would be more appropriate.

R) We have further stressed the difference between the potential values and the realized values of kelp forests.

L202

The potential value generated by fisheries species occupying kelp forests ...

Line 376. Were the regional costs of nitrogen removal used, or a global average?

S) We have added text to reflect that we used the average of several nitrogen removal schemes around the world

L471-472

We collected market prices for the social cost of carbon and averaged nutrient trading schemes from around the world (Appendix 9).

Line 385. Is there a reference for this approach?

T) We have added an appropriate reference.

Žižlavský, O. (2014). Net present value approach: method for economic assessment of innovation projects. *Procedia-Social and Behavioral Sciences*, 156, 506-512.

Appendices: the Scheibling et al. 1999 reference for sea urchin densities is dated. There are currently little to no sea urchins in this region due to disease, and the sea urchin fisheries closed in the last decade from lack of supply. Disease as a control of sea urchin populations in Nova Scotian kelp beds. CJ Feehan, RE Scheibling. *Marine Ecology Progress Series* 500, 149-158. The values used are for NFL, a different region, which is largely in a barrens state (Work by P. Gagnon, Memorial University).

U) We have removed this data point from our analysis

Reviewer #1 (Remarks to the Author):

The authors have done a tremendous job revising and addressing the reviewers' comments. They've satisfied my prior concerns beautifully by 1) expanding the number of genera and datasets included in the analysis, 2) adding a sensitivity analysis to rule out the effect of geographically biased datasets, 3) evaluating whether a different valuation system would change the results for carbon, 4) providing further justification or caveats where necessary, and other revisions. This is a very strong paper that is poised to make a significant contribution to fields of natural capital and blue carbon research. I am confident this paper will also be of broad interest to the readership of Nature Communications.

Reviewer #3 (Remarks to the Author):

The paper works to estimate a global ecosystem services value for macroalgae. They focus on fisheries and carbon and nutrient uptake values as the main ecosystem services. These would be the first estimates of a global value per hectare per year. The order of magnitude is several times higher than other estimates scaled to the global level would provide.

This set of numbers, if published, can be expected to be used frequently and widely relied upon for all sorts of arguments concerning how and where to try to intervene, often with public money, in ecological processes and marine resource markets. Thus whether this large difference is based on underestimates due to e.g. neglect of the topic or imperfect methods, or rather reflects substantial overestimation or a (different) imperfect method bears considerable scrutiny. The authors themselves have done a creditable job of trying to address this concern and I appreciate the effort to make what they have done fairly transparent. They argue that their estimates are the "potential economic value as opposed to the realized value" and that this is the main reason why their figures are higher than others.

Still, I am concerned that there is something of a misunderstanding about the potential value of the ecosystem and its dynamics that does in fact make these values higher than we should accept, for the three sets of ecosystem services included (fisheries, nutrient cycling, and carbon removal). This is not to say that there aren't missing services, e.g. to recreation, coastal erosion, etc., that would contribute substantially to the magnitude of value if they could be included – but these cannot be excuses for using unrealistic numbers for the services we can value.

The authors first introduce their uses of potential value in lines 85-87. Here, it reads as if the authors equate potential economic value with the value of natural capital and realized economic value as the value of human derived capital. This is not correct – human derived capital is the capital used to harness the service flows from the ecosystem capital stock (e.g. the fishing gear, vessels, etc), so that the human derived capital determines the realized value rather than is that value.

While I suspect that this is understood by the authors and is simply somewhat poor wording, it seems relevant to take the opportunity to reflect on the challenges of interacting stocks and flows in any valuation process. The kelp ecosystems are a stock of natural capital, from which a variety of production processes flow. Some of these production processes could be considered "within" the natural systems while others are harnessed by human-derived capital and brought into anthropocentric realms, making them easier to assess in terms of monetary values. What the authors are trying to do is to calculate the realized value of these latter flows, and then extrapolate them to put monetary values on the production processes that are potentially available but not yet quantifiably used in human systems. This would create a defensible set of estimates of the potential value of the flows from the kelp systems.

There seem to be a couple of stumbling blocks to successfully doing this, with some more easily fixed than others.

1. There is a conflation of potential value per year (flow) and potential value of the system (stock).

In lines 235-238 it seems that the estimates for potential value are calculated as if the total production were removed and placed on the market. This would be correct if the assessment pertained to a one-time conversion of the asset into another form (e.g. money in the bank from the harvest of all the species). It is not correct as a per-year assessment because as the authors note, this would deplete the animal biomass and there would be no future fishery. If the goal is not assessing a one-time conversion, then the value that should be used for potential value must be for sustainable production of the fisheries – that is, something like estimated maximum sustainable yield (or preferably maximum economic yield). This point is made in the cited de Groot et al article as well. This is particularly needed as the calculations for Net Present Value attribute this potential value on an annual basis and add it in each year for twenty years, creating significant overcounting (relative to the interest accrued from having the converted asset in a bank account, that could be added in each year after conversion, for example).

2. The connection between the kelp ecosystems and the fisheries productivity is tenuous

Without a clearer connection here, the fisheries portion of the values assessment is pretty speculative from the start. I am not convinced that the caveat in lines 219-223 is enough to conquer the problem. The 1358 surveys that are meant to inform about the connection between kelp species and fisheries production generate an impressive 39,071 observations of site X species X date with calculations of species density (n) per square meter (Appendix 5). At least some cases of the same species being counted at the same location on different dates exist in the data (e.g. the first entries for *Astrarium haematragum* in Japan in the Akimoto et al study give three measures of N/m^2 over the course of about a year), which theoretically might make it possible to explain statistical relationships between N/m^2 and kelp conditions. (one side question – while the same sites are surveyed multiple times, as with this first study – the recorded N/m^2 is never zero, while I am assuming that e.g. *Calliostoma unicum* is not present 2 of the 3 dates but is present and recorded once. Without the zero cases included, it seems like the species-level estimates will be biased?)

However, the records of the studies provided here do not give any information about levels (other than the main kelp genera & species) or shifts in the kelp conditions at the site, so that in spite of this finely grained data, there is no straightforward way to connect the kelp to the fisheries productivity within the sites, and so no way to gage the dependency of the fisheries on the kelp or to assess marginal changes. The date and lat-long are recorded; I suppose it could be possible to reconstruct some sort of light index for the place-date combinations and estimate an effect on N/m^2 , but there would be no accounting for actual kelp conditions and this effort is unlikely to yield much of use.

Still, spatial considerations could perhaps be used more effectively, given the differences in estimates in lines 114-120. I can see from a quick look at the data that for the two genera that are present in both northern and southern latitudes, *Macrocystis* is associated with a statistically significantly higher N/m^2 in the south while *Ecklonia* is associated with a statistically significantly higher N/m^2 in the north (and *Ecklonia* is statistically significantly related to higher N/m^2 in the more diverse north than the *E. radiata*-dominated south, vice-versa for *Macrocystis*). I am not well enough versed in all the global kelp-fisheries-markets combinations to know how this pans out in the existing analysis with respect to value estimates or ecosystem roles, but the combined ecological and economic implications of the spatial divisions may be one worth pursuing further to create some improved understanding of marginal changes in kelp conditions and associated fisheries productivity.

3. The role of markets is overly simplistic and not integrated into the valuation process as needed

The paper boils down markets to snapshots of prices. 553 (once NA cases are removed from my running of the R script creating 'costs_used.csv' at least) prices for individual

species are collected and converted to 2020 USD (side note: the specific exchange rates and CPI choices used should be more clearly documented) and then matched back to the site X species information, but these prices come from a diverse set of market conditions and institutional settings which may significantly impact prices, and are certainly not directly comparable to one another.

For example: The level of processing varies, with about 1/3 of the prices attributed to some sort of "live" product while others are processed all the way through to fillets. Live products in general command a higher price in fisheries; here this is no exception, with the average 'live' price being about 3 times as high as the more processed values. On the other hand, they will have a much shorter shelf life – so this additional value is more easily lost, and markets are unlikely to be able to absorb increased output at these higher prices. Thus bringing more live product to market would result in faster price declines than bringing more processed product.

In markets for the same or similarly purposed species, it might be possible to estimate price elasticities (the percent change in quantity demanded from a percent change in price) even across product types (live/frozen), and these price elasticities could be used to help estimate shadow prices for changes in expected value if realized productivity were increased, and so facilitate pricing the potential ecosystem. In this case, however, there is an additional complication that the live species appear to mainly be part of the ornamental trade, not easily comparable to commercial/industrial fisheries. (side note: given that ornamentals make up over 40% of the species with prices, the ornamental trade should be explicitly mentioned in the paper. Given the method described in 442-453, it is a little unclear if ornamental trade prices might have been used for species that are not ornamentals but are in genera or families (or order or phyla even?!) that are, and were the 'closest' value – if this is the case, I'd argue end market should determine proximity rather than classification).

More importantly for this research, the differences in markets highlight an essential part of valuation that is not addressed here at all: market structures matter for valuation. In fisheries and marine resource exploitation in general, there is for example a great deal of value that could be generated not from increased harvest from the sea but from e.g. improved regulation and governance. The ornamental trade is a clear case in point: the mortality rates associated with ornamental fish trade are extremely high throughout the supply chain, and they vary with any number of factors including the method of harvest, distance to market, incentives of suppliers, and the know-how of the end consumer. Any number of regulatory or market innovations can be considered (and are, in localized environments and/or certain supply chains) that can reduce these mortality figures, thereby increasing the value of existing levels of harvests. This means, unfortunately, that estimates of \$/kg/ha/yr are dependent on market and institutional conditions that cannot easily be abstracted away from to create the sort of meaningful global estimate the authors seek. Fenichel and Abbott (2014) and related work may help in explaining this problem more completely as well as in providing solutions.

A direct consequence of oversimplifying the market analyses is that in addition to overall miscalculation of the total values, the relative shares in Fig 2, which are based on the estimated values of the species scaled for their presence, are not likely to hold either in a static assessment of more accurate shadow prices or in a dynamic assessment of impacts from ecosystem changes.

There are fewer challenges with the nutrient cycling and carbon sequestration calculations, in part because the connection between the kelp production and these services is more directly estimable but also through acknowledgements such as: lines 268-274 and around the SCC (lines 295-299) that there are any number of factors that will influence the relationships that are not accounted for. At the very least, a similar effort to consider sensitivities in prices as a function of the points made above could assist in contextualizing the estimates more fully. Lines 314-320, where readers are explicitly reminded that the differences in initial

conditions (here, of elevated nutrient levels) impact the valuations and should guide their use (or omission of use).

Small issues:

The appendices are not called entirely in order in the text.

The title for figure 1 is "fisheries production economic value" but the Y-axis appears to be biomass.

Line 172 you need to include the discount rate used in introducing the NPV.

Line 317 reads funny – I do not think you mean that re-oxygenation is a carry-on effect of pollution, but rather the opposite.

Line 350 – missing a zero

References:

Fenichel, E. P., & Abbott, J. K. (2014). Natural capital: from metaphor to measurement. *Journal of the Association of Environmental and Resource Economists*, 1(1/2), 1-27.

REVIEWER COMMENTS

Reviewer #1 (Remarks to the Author):

The authors have done a tremendous job revising and addressing the reviewers' comments. They've satisfied my prior concerns beautifully by 1) expanding the number of genera and datasets included in the analysis, 2) adding a sensitivity analysis to rule out the effect of geographically biased datasets, 3) evaluating whether a different valuation system would change the results for carbon, 4) providing further justification or caveats where necessary, and other revisions. This is a very strong paper that is poised to make a significant contribution to fields of natural capital and blue carbon research. I am confident this paper will also be of broad interest to the readership of Nature Communications.

We thank reviewer #1 for their insights and contributions to our manuscript and are pleased that they are satisfied with the completed revisions.

Reviewer #3 (Remarks to the Author):

We thank the reviewers for their constructive feedback, particularly with respect to the economic analysis. They have raised valid points, which upon addressing, we believe have increased the quality of the work. We have adjusted for differences in the market values of fishery species around the world by adjusting our estimates for purchasing power, applying a discount cost of capital rate based on the country the fishery is located, and applying a supply chain discount rate which accounts for differences in the processing level of the species. Further, we have applied a harvest rate to the fisheries biomass so that it now represents the amount of biomass that would be reasonably harvested in a year. We also explored how changing this rate changes the results of our work. We have also explored the relationship between kelp condition (e.g., cover or density) and fisheries biomass and incorporated that into our discussion. These changes are reflected throughout the document along with other considerations and clarifications.

We provide our individual responses to the reviewer below and have denoted the sections as follows

Red: Reviewer's comment

Blue: Our response

Black: In text changes made

The paper works to estimate a global ecosystem services value for macroalgae. They focus on fisheries and carbon and nutrient uptake values as the main ecosystem services. These would be the first estimates of a global value per hectare per year. The order of magnitude is several times higher than other estimates scaled to the global level would provide.

This set of numbers, if published, can be expected to be used frequently and widely relied upon for all sorts of arguments concerning how and where to try to intervene, often with public money, in ecological processes and marine resource markets. Thus whether this large difference is based on underestimates due to e.g. neglect of the topic or imperfect methods, or rather reflects substantial overestimation or a (different) imperfect method bears considerable scrutiny. The authors themselves have done a creditable job of trying to address this concern and I appreciate the effort to make what they have done fairly transparent. They argue that their estimates are the “potential economic value as opposed to the realized value” and that this is the main reason why their figures are higher than others.

Still, I am concerned that there is something of a misunderstanding about the potential value of the ecosystem and its dynamics that does in fact make these values higher than we should accept, for the three sets of ecosystem services included (fisheries, nutrient cycling, and carbon removal). This is not to say that there aren't missing services, e.g. to recreation, coastal erosion, etc., that would contribute substantially to the magnitude of value if they could be included – but these cannot be excuses for using unrealistic numbers for the services we can value.

The authors first introduce their uses of potential value in lines 85-87. Here, it reads as if the authors equate potential economic value with the value of natural capital and realized economic value as the value of human derived capital. This is not correct – human derived capital is the capital used to harness the service flows from the ecosystem capital stock (e.g.

the fishing gear, vessels, etc), so that the human derived capital determines the realized value rather than is that value.

We have reworded this section and removed the phrase “human derived capital” as it was indeed confusing. Instead, we focus on the concept of stocks and flows as follows:

L88-90

As a result, our work describes the capacity of global kelp forests’ to supply ecosystem services³¹. This capacity is the potential economic value (herein value) as opposed to the realized value.

While I suspect that this is understood by the authors and is simply somewhat poor wording, it seems relevant to take the opportunity to reflect on the challenges of interacting stocks and flows in any valuation process. The kelp ecosystems are a stock of natural capital, from which a variety of production processes flow. Some of these production processes could be considered “within” the natural systems while others are harnessed by human-derived capital and brought into anthropocentric realms, making them easier to assess in terms of monetary values. What the authors are trying to do is to calculate the realized value of these latter flows, and then extrapolate them to put monetary values on the production processes that are potentially available but not yet quantifiably used in human systems. This would create a defensible set of estimates of the potential value of the flows from the kelp systems. There seem to be a couple of stumbling blocks to successfully doing this, with some more easily fixed than others.

1. There is a conflagration of potential value per year (flow) and potential value of the system (stock).

In lines 235-238 it seems that the estimates for potential value are calculated as if the total production were removed and placed on the market. This would be correct if the assessment pertained to a one-time conversion of the asset into another form (e.g. money in the bank from the harvest of all the species). It is not correct as a per-year assessment because as the authors note, this would deplete the animal biomass and there would be no future fishery. If

the goal is not assessing a one-time conversion, then the value that should be used for potential value must be for sustainable production of the fisheries – that is, something like estimated maximum sustainable yield (or preferably maximum economic yield). This point is made in the cited de Groot et al article as well. This is particularly needed as the calculations for Net Present Value attribute this potential value on an annual basis and add it in each year for twenty years, creating significant overcounting (relative to the interest accrued from having the converted asset in a bank account, that could be added in each year after conversion, for example).

We appreciate the reviewer's point and agree that using a harvest rate/sustainable yield value is more appropriate for a per year/rate value. Therefore, we have applied a range of harvest rates from 20 – 70%, and used a reported average value of 38% to report the total ecosystem service values. These changes are reflected in the methods and results below.

L479-484

To ensure a future harvest, not all fish production is harvested in one year. As a result, there is considerable variation in reported sustainable harvest rates for fisheries^{40,41}. Therefore, in our economic evaluation, we considered that a range from 20 – 70% of production is harvested each year while using an observed average value of 38⁴¹% as a base rate. The sustainable harvest level will vary by species, region, and time but these numbers cover the span of observed values.

L246-251

For our economic evaluation, we aimed to value the sustainable harvestable fisheries biomass that is produced each year^{55,56}. We chose this value over the total biomass produced to not promote the complete extraction of fisheries biomass and to enable the economic evaluation for consecutive years as opposed to a single year value (i.e., if all the biomass is removed in one year, there is no value left for the second year). Another alternative would be to report the realized value, i.e., the amount that is extracted, sold, and recorded by fisheries agencies.

L262-264

The harvest rate will influence the realized economic value and what is sustainable will vary by species, region, and even year. Therefore, the harvest rates we used are only for illustrative purposes and should not be used to set fishing policy.

L123-126

Using our selected harvest ranges, 20 and 70%, the range of economic values were *Ecklonia* (22,800\$ - \$79,800), *Laminaria/Saccharina* (\$19,500 – \$68,300), *Lessonia* (\$6,650 – \$23,300), *Macrocystis* (\$15,300 – \$53,500), and *Nereocystis* (\$33,500) (Appendix 3).

L219-223

The average economic value of that 38% harvest is \$35,220 per year, while a 20% harvest yields \$18,537 a year and a 70% harvest yields \$64,882 a year. Under these same scenarios, the global value of kelp forests shifts from \$523 billion to \$469 billion in the low harvest scenario and to \$602 billion in the high harvest scenario.

2. The connection between the kelp ecosystems and the fisheries productivity is tenuous

Without a clearer connection here, the fisheries portion of the values assessment is pretty speculative from the start. I am not convinced that the caveat in lines 219-223 is enough to conquer the problem. The 1358 surveys that are meant to inform about the connection between kelp species and fisheries production generate an impressive 39,071 observations of site X species X date with calculations of species density (n) per square meter (Appendix 5).

At least some cases of the same species being counted at the same location on different dates exist in the data (e.g. the first entries for *Astraliu haematragum* in Japan in the Akimoto et al study give three measures of N/m² over the course of about a year), which theoretically might make it possible to explain statistical relationships between N/m² and kelp conditions. (one side question – while the same sites are surveyed multiple times, as with this first study – the recorded N/m² is never zero, while I am assuming that e.g. *Calliostoma unicum* is not present 2 of the 3 dates but is present and recorded once. Without the zero cases included, it seems like the species-level estimates will be biased?)

To answer the side question. Each observation is considered as its own data point, species densities were not averaged across time or location. Therefore we do not need to add zero values as site averages are not being calculated.

However, the records of the studies provided here do not give any information about levels (other than the main kelp genera & species) or shifts in the kelp conditions at the site, so that in spite of this finely grained data, there is no straightforward way to connect the kelp to the fisheries productivity within the sites, and so no way to gage the dependency of the fisheries on the kelp or to assess marginal changes. The date and lat-long are recorded; I suppose it could be possible to reconstruct some sort of light index for the place-date combinations and estimate an effect on N/m², but there would be no accounting for actual kelp conditions and this effort is unlikely to yield much of use.

Still, spatial considerations could perhaps be used more effectively, given the differences in estimates in lines 114-120. I can see from a quick look at the data that for the two genera that are present in both northern and southern latitudes, *Macrocystis* is associated with a statistically significantly higher N/m² in the south while *Ecklonia* is associated with a statistically significantly higher N/m² in the north (and *Ecklonia* is statistically significantly related to higher N/m² in the more diverse north than the *E. radiata*-dominated south, vice-versa for *Macrocystis*). I am not well enough versed in all the global kelp-fisheries-markets combinations to know how this pans out in the existing analysis with respect to value estimates or ecosystem roles, but the combined ecological and economic implications of the

spatial divisions may be one worth pursuing further to create some improved understanding of marginal changes in kelp conditions and associated fisheries productivity.

The relationship between kelp condition and ecosystem services is an important one. However, this research intended to present a range and average of observed values, not to predict services provided based on environmental or biotic conditions. While we agree it is an important question, we believe running a full analysis on this is a separate research question.

Nevertheless, we have used our available data to explore this question to some degree by analysing the relationship between kelp forest density and fisheries biomass for a subset of data from which we had these values (Appendix 6). We also point to past research which demonstrates a positive link between kelp forests and fishery production to expand on this link and we highlight that understanding the exact contribution of kelp forest habitat to fisheries remains an important research gap.

L239-245

The exact contribution of kelp forest habitat to these fisheries services remains an important next step in understanding how kelp forests support food webs⁵⁴. Our analysis of the relationship between kelp forest density and fisheries biomass revealed that there was a positive relationship between kelp density and fisheries biomass (Appendix 6). A more detailed review paper¹⁵ revealed that kelp forests had a positive effect on fish abundance in 19 of the 24 studies reviewed, a positive effect on crustacean abundance in 4/4 studies, and a positive effect on gastropod abundance in 2/3 studies.

L515-519

Kelp forest density was not associated with all our fishery survey data, so we ran a limited analysis of the relationship between kelp density and fisheries biomass.

Together, we had 91 observations from 47 independent sites. We used a mixed effect model with “site” as a random factor to account for multiple observations at the same location but at different dates¹¹⁰.

3. The role of markets is overly simplistic and not integrated into the valuation process as needed

The paper boils down markets to snapshots of prices. 553 (once NA cases are removed from my running of the R script creating ‘costs_used.csv’ at least) prices for individual species are collected and converted to 2020 USD (side note: the specific exchange rates and CPI choices used should be more clearly documented) and then matched back to the site X species information, but these prices come from a diverse set of market conditions and institutional settings which may significantly impact prices, and are certainly not directly comparable to one another.

We appreciate the reviewer’s insights into how to adjust our economic values to better reflect market conditions and have worked to address the individual points as detailed in response to additional points below, by including discount rates, adjusting prices for purchasing power parity and accounting for cost of capital. We also note that there is a precedent in aggregating biophysical and economic data at the global level and making comparisons between those values. We acknowledge the approach requires assumptions but think the work is still worthwhile.

We now clearly document the exchange rates and CPI index used in the data in Appendix 11.

We also discuss how market prices will affect the outcomes of this work.

L379-383

Market prices for the fish species will depend on the year, season, level of processing, distance to market, risk of spoilage, and other factors such as changes in regulation and governance⁸⁹⁻⁹¹. Similarly, the price of carbon, nitrogen, and phosphorus will also change through time. As the market prices change, there will be corresponding changes to the estimated values presented here and these values are thus a snapshot.

For example: The level of processing varies, with about 1/3 of the prices attributed to some sort of “live” product while others are processed all the way through to fillets. Live products in general command a higher price in fisheries; here this is no exception, with the average ‘live’ price being about 3 times as high as the more processed values. On the other hand, they will have a much shorter shelf life – so this additional value is more easily lost, and markets are unlikely to be able to absorb increased output at these higher prices. Thus, bringing more live product to market would result in faster price declines than bringing more processed product.

We have applied a discount rate of 2.5% if a price was given for a highly processed state (e.g., dried, fillet, or roe) and an additional 2.5% discount rate if the price was given for a “high risk” state which may be spoiled quickly (e.g., live, roe, or fresh). This adjustment is discussed in the methods and numbers are given in Appendix 12. We chose the value to situate it near to the other discount values applied (3% discount rate, 3-15% cost of capital) and not have an outsized influence on the result. We acknowledge that this only partially addresses the issue and have highlighted that as well.

L505-513

Further, as the prices were obtained for products with different levels of processing (e.g. live versus filleted versus dried), we adjusted for the resources required for each processing type as well as the risk of that product spoiling and being worth nothing. The discount rate for a highly processed product or a likely to spoil product was 2.5%, therefore a maximum discount rate of 5 % per price was applied (Appendix 13). These values were approximated to help account for these differences but do not fully address this issue and may be improved upon in future analysis.

In markets for the same or similarly purposed species, it might be possible to estimate price elasticities (the percent change in quantity demanded from a percent change in price) even across product types (live/frozen), and these price elasticities could be used to help estimate shadow prices for changes in expected value if realized productivity were increased, and so facilitate pricing the potential ecosystem. In this case, however, there is an additional

complication that the live species appear to mainly be part of the ornamental trade, not easily comparable to commercial/industrial fisheries. (side note: given that ornamentals make up over 40% of the species with prices, the ornamental trade should be explicitly mentioned in the paper. Given the method described in 442-453, it is a little unclear if ornamental trade prices might have been used for species that are not ornamentals but are in genera or families (or order or phyla even?!) that are, and were the 'closest' value – if this is the case, I'd argue end market should determine proximity rather than classification).

We understand the desire to calculate the marginal prices with increased supply or demand but do not think this analysis is feasible given the number of species we consider. Further, these analyses will always be a snapshot and the market prices will vary from when we collected the data to when we submitted to when it was reviewed. We have noted the year that all the prices were collected for in Appendix 11 but think this will be an inevitable limitation of our study.

We have now explicitly mentioned that ornamental species are included in our analysis.

L493-495

Species market values were recorded at differing levels of processing (e.g., dried versus alive) and some were sold for consumption while others were sold on the ornamental market. All values are recorded in the supplement (Appendix 11).

We have also clarified that species were only ever aggregated at the genus level. Prior to the analysis we thought more coarse groups might be required but that was not the case.

L474-477

If a species had no length or weight-length relationship values, we used values from species in the same genera or family. If there was no value available in the same genus or family, we searched for biomass estimates.

More importantly for this research, the differences in markets highlight an essential part of valuation that is not addressed here at all: market structures matter for valuation. In fisheries and marine resource exploitation in general, there is for example a great deal of value that could be generated not from increased harvest from the sea but from e.g. improved regulation

and governance. The ornamental trade is a clear case in point: the mortality rates associated with ornamental fish trade are extremely high throughout the supply chain, and they vary with any number of factors including the method of harvest, distance to market, incentives of suppliers, and the know-how of the end consumer. Any number of regulatory or market innovations can be considered (and are, in localized environments and/or certain supply chains) that can reduce these mortality figures, thereby increasing the value of existing levels of harvests. This means, unfortunately, that estimates of \$/kg/ha/yr are dependent on market and institutional conditions that cannot easily be abstracted away from to create the sort of meaningful global estimate the authors seek. Fenichel and Abbott (2014) and related work may help in explaining this problem more completely as well as in providing solutions.

The reviewer makes an important point about differing market conditions and we have taken two steps to address this issue. First, we have adjusted all the prices for purchasing power parity.

L546-547

All dollar values in our analysis are presented in international dollars for the year 2020 and have been adjusted using the purchasing power exchange rate¹¹⁵, unless stated otherwise.

L495-497

The fisheries values were then adjusted for purchasing power and converted into international dollars/ Kg¹³ and adjusted for inflation to the year 2020 (Figure 5).

Second, we have accounted for the different costs of capital in each country.

L502-505

Because the amount of money invested before turning a profit varies by countries, we accounted for this “cost of capital” based on the country the fish was extracted from. These values ranged from 3 – 15% (Appendix 12)^{66,108,109}.

We also acknowledge how differences in these prices will alter the results of our study.

L379-381

Market prices for the fish species will depend on the year, season, level of processing, distance to market, risk of spoilage, and other factors such as changes in regulation and governance⁸⁹⁻⁹¹.

L382-383

As the market prices change, there will be corresponding changes to the estimated values presented here.

A direct consequence of oversimplifying the market analyses is that in addition to overall miscalculation of the total values, the relative shares in Fig 2, which are based on the estimated values of the species scaled for their presence, are not likely to hold either in a static assessment of more accurate shadow prices or in a dynamic assessment of impacts from ecosystem changes.

We hope that the steps detailed above (inclusion of discount rates, adjustment of prices for purchasing power parity, accounting for cost of capital) contribute to a better contextualised estimation of the species' prices and mitigate this concern.

There are fewer challenges with the nutrient cycling and carbon sequestration calculations, in part because the connection between the kelp production and these services is more directly estimable but also through acknowledgements such as:

lines 268-274 and around the SCC (lines 295-299) that there are any number of factors that will influence the relationships that are not accounted for. At the very least, a similar effort to consider sensitivities in prices as a function of the points made above could assist in contextualizing the estimates more fully.

With regards to SCC (social cost of carbon), we have discussed how fluctuations in the price of carbon may impact its value in the future.

L317-324

The mean economic value of carbon capture was only \$163 per hectare per year even though we used the social cost of carbon (~\$45/ton C⁷⁵), a relatively high estimate that incorporates the social and environmental externalities of increased atmospheric CO₂ concentrations in our evaluation. Previous work suggests that even the social cost of carbon underestimates the true value of carbon capture⁷⁶. Nevertheless, even if the price of carbon were to increase ten-fold to \$450/ton, the resulting economic value of carbon capture in kelp forests would remain relatively low at \$1,630/ hectare/year.

Because it was such a small economic factor in our analysis we do not think a second, more in depth sensitivity analysis is justified in this instance.

Lines 314-320, where readers are explicitly reminded that the differences in initial conditions (here, of elevated nutrient levels) impact the valuations and should guide their use (or omission of use).

We have further stressed how the context for evaluations influences the price as follows.

L333-340

Placing an economic value on the nitrogen removed from the ocean requires some simplification. First, we obtained estimates of nutrient trading schemes from the Eastern United States, Southern Australia, and Europe. These schemes are based on the replacement cost of the service, that is, how much it would cost to build a water treatment plant to remove the same amount of nitrogen as the kelp. Our approach equates the ocean-based removal of these nutrients with these numbers. While there are inherent mechanistic differences between upstream and ocean-based removal, these equivalencies are necessary in the absence of market-based values for these processes⁷⁷.

Small issues:

The appendices are not called entirely in order in the text.

We have ensured the appendices are now in the correct order

The title for figure 1 is “fisheries production economic value” but the Y-axis appears to be biomass.

We have corrected the Y-axis of figure 1.

Line 172 you need to include the discount rate used in introducing the NPV.

We have noted the 3% discount rate we used

L558-560

The net present value was calculated using a 3% discount rate^{115,116} and represents the current present value of 20 years of services provided by 1 hectare of kelp forest (i.e., potential economic value from 2021 – 2041)¹¹⁷.

Line 317 reads funny – I do not think you mean that re-oxygenation is a carry-on effect of pollution, but rather the opposite.

We have reworded this sentence to clarify as follows:

L348-351

Conversely, this value may also increase as kelp forests in these zones would provide additional services and value by reoxygenating hypoxic zones that are often caused by nutrient pollution⁷⁹ and we have not included that. Further incorporating these complexities would increase the accuracy of our evaluations.

Line 350 – missing a zero

We have added the zero

References:

Fenichel, E. P., & Abbott, J. K. (2014). Natural capital: from metaphor to measurement. *Journal of the Association of Environmental and Resource Economists*, 1(1/2), 1-27.

Thank you, we have read this reference and incorporated it into our response.

Reviewer #1 (Remarks to the Author):

This manuscript represents the first global assessment of the ecosystem service value of kelp forests, focusing on six major habitat forming kelp genera (i.e., Ecklonia, Laminaria, Lessonia, Macrocystis, Nereocystis, and Saccharina) and three main ecosystem services (i.e., fisheries production, nutrient removal, and carbon sequestration). The authors explore patterns of global variation in ecosystem service value across regions and genera, using a database of site level area-specific estimates for each service derived from variety of sources. The results demonstrate the considerable ecological and economic value of kelp forests, even showing a 3.25-fold increase in monetary value over previous studies. Importantly, the authors find that the economic value of kelps is mainly driven by fisheries production and nitrate removal services rather than blue carbon storage, suggesting the need to use caution when promoting carbon capture as a purely economic incentive for protecting or restoring kelps.

Reviewer #3 had raised several valid concerns—most of which are now addressed through extensive modification to the text and analyses. In particular, the authors have since taken steps to 1) clarify the language around the potential and realized economic value of services, 2) better quantify the link between kelp forest ecosystems and fishery production while acknowledging nuances and the limitations of the study, 3) better address market factors affecting service value (ie., discount rates, purchasing power, etc) among others. Considering these new changes, in addition to the substantial changes made during the last round of reviews, I consider this paper to be much improved. This is a strong paper poised to make an important contribution to fields of ecosystem service science, ocean accounting, blue carbon policy and markets, and natural climate solution.

Reviewer #3 (Remarks to the Author):

In general the revisions are extremely helpful and well done. I am still concerned, however, about the tenuous connection between kelp and biomass productivity. In lines 241-243 the authors state that the data shows a positive relationship between kelp density and fisheries biomass shown in Appendix 6. Unfortunately, Appendix 6 does not establish a statistically significant relationship between kelp density and fisheries biomass. There is a positive coefficient, but it is statistically insignificant and cannot therefore accurately be considered any differently than no relationship, or even a negative relationship. The claim that the kelp forests generate all this value (lines 217-218) is not accurate because you cannot attribute all that biomass to the kelp. (Even if there is a positive statistical relationship – the entire value cannot be attributed to the kelp (at least with the current specification) – what would the fisheries production be with no kelp? From the linear regression you have here, for example, the intercept suggests that 111.9 g/m²/year are totally independent of the presence of kelp, and at a kelp density of 20 n/m² you'd only get another 100 g/m²/year of additional biomass – so only about half – at the max kelp density shown – can be (perhaps) attributed to kelp. The attribution of value needs to be adjusted for this.

The regression is very much at the macro scale. If there are enough datapoints, which with 91 observations it looks like there could be but it will be close, it would be worth controlling for the species biomasses mentioned in the paragraph above – just in a very general division of 'transitory' (and not then expected to matter much), 'life stage relationship present' or 'obligate species' (I would interact type and biomass so that the slopes of the relationships can vary rather than just dummies for the type, which will only shift the intercept). I would expect that this could confirm that esp. for obligate species, there is a significant positive relationship. If all of the relationships are confirmed (it is possible they all will be, just with different slopes) then I think you can continue to confidently use the full biomass in the economic calculations – but if it is only the obligate for example, it would be best to scale the economics by the biomass of

species types that do seem to have this relationship in a confirmed way. Additionally, using the information in the Bertocci et al paper (and/or the other literature indicated in lines 235-236, you can argue that the species biomass for the species they evaluate can be considered to have a significant positive relationship and be included in the value (assuming there is sufficient such evidence in the paper, which I do not have).

Without statistical confirmation of the positive relationship you could probably use some Monte Carlo simulations with the data to generate draws of the potential relationship and generate some average biomass (and therefore value) scaled by the probability of its dependence on the kelp, with estimated variance parameters that informs the sensitivity analysis directly. There may be some other clever way to deal with this uncertain relationship – perhaps along the lines of what is described in lines 471ff on dealing with uncertainty in the biomass itself or lines 505ff dealing with different product types.

I also noticed somewhat randomly that Appendix 3 column L seems to duplicate into column M at row 59, shifting the rest of the data set to the right. If this did not happen after all analytical use of these data, such use should probably be checked.

REVIEWER COMMENTS

We thank the reviewers for their continued consideration of our work and provide our individual responses to the reviewer below. We have denoted the sections as follows:

Red: Reviewer's comment

Blue: Our response

Black: In text changes

Reviewer #1 (Remarks to the Author):

This manuscript represents the first global assessment of the ecosystem service value of kelp forests, focusing on six major habitat forming kelp genera (i.e., Ecklonia, Laminaria, Lessonia, Macrocystis, Nereocystis, and Saccharina) and three main ecosystem services (i.e., fisheries production, nutrient removal, and carbon sequestration). The authors explore patterns of global variation in ecosystem service value across regions and genera, using a database of site level area-specific estimates for each service derived from variety of sources. The results demonstrate the considerable ecological and economic value of kelp forests, even showing a 3.25-fold increase in monetary value over previous studies. Importantly, the authors find that the economic value of kelps is mainly driven by fisheries production and nitrate removal services rather than blue carbon storage, suggesting the need to use caution when promoting carbon capture as a purely economic incentive for protecting or restoring kelps.

Reviewer #3 had raised several valid concerns—most of which are now addressed through extensive modification to the text and analyses. In particular, the authors have since taken steps to 1) clarify the language around the potential and realized economic value of services, 2) better quantify the link between kelp forest ecosystems and fishery production while acknowledging nuances and the limitations of the study, 3) better address market factors affecting service value (ie., discount rates, purchasing power, etc) among others. Considering these new changes, in addition to the substantial changes made during the last round of reviews, I consider this paper to be much improved. This is a strong paper poised to make an important contribution to fields of ecosystem service science, ocean accounting, blue carbon policy and markets, and natural climate solution.

We appreciate the reviewer's continued consideration of our work and thank them for the constructive remarks which improved the research.

Reviewer #3 (Remarks to the Author):

In general the revisions are extremely helpful and well done. I am still concerned, however, about the tenuous connection between kelp and biomass productivity.

In lines 241-243 the authors state that the data shows a positive relationship between kelp density and fisheries biomass shown in Appendix 6. Unfortunately, Appendix 6 does not establish a statistically significant relationship between kelp density and fisheries biomass. There is a positive coefficient, but it is statistically insignificant and cannot therefore accurately be considered any differently than no relationship, or even a negative relationship. The claim that the kelp forests generate all this value (lines 217-218) is not accurate because you cannot attribute all that biomass to the kelp. (Even if there is a positive statistical relationship – the entire value cannot be attributed to the kelp (at least with the current specification) – what would the fisheries production be with no kelp? From the linear regression you have here, for example, the intercept suggests that 111.9 g/m²/year are totally independent of the presence of kelp, and at a kelp density of 20 n/m² you'd only get another 100 g/m²/year of additional biomass – so only about half – at the max kelp density shown – can be (perhaps) attributed to kelp. The attribution of value needs to be adjusted for this.

The regression is very much at the macro scale. If there are enough datapoints, which with 91 observations it looks like there could be but it will be close, it would be worth controlling for the species biomasses mentioned in the paragraph above – just in a very general division of 'transitory' (and not then expected to matter much), 'life stage relationship present' or 'obligate species' (I would interact type and biomass so that the slopes of the relationships can vary rather than just dummies for the type, which will only shift the intercept). I would expect that this could confirm that esp. for obligate species, there is a significant positive relationship. If all of the relationships are confirmed (it is possible they all will be, just with different slopes) then I think you can continue to confidently use the full biomass in the economic calculations – but if it is only the obligate for example, it would be best to scale the economics by the biomass of species types that do seem to have this relationship in a confirmed way. Additionally, using the information in the Bertocci et al paper (and/or the other literature indicated in lines 235-236, you can argue that the species biomass for the species they evaluate can be considered to have a

significant positive relationship and be included in the value (assuming there is sufficient such evidence in the paper, which I do not have).

Without statistical confirmation of the positive relationship you could probably use some Monte Carlo simulations with the data to generate draws of the potential relationship and generate some average biomass (and therefore value) scaled by the probability of its dependence on the kelp, with estimated variance parameters that informs the sensitivity analysis directly. There may be some other clever way to deal with this uncertain relationship – perhaps along the lines of what is described in lines 471ff on dealing with uncertainty in the biomass itself or lines 505ff dealing with different product types.

We thank reviewer 3's consideration of our research and understand the distinction they are expressing. How much an organism depends on a habitat is an important research question and an important consideration when expressing the economic value of these organism.

While we previously worked to describe the value present in kelp forests, we appreciate the desire to consider the gradient of that dependency. Therefore, we have worked to classify 187 genera as having a high, medium, low, or no dependency on kelp forests and adjusted the economic evaluation using partial dependency values (66%, 33%, 0%) for the medium, low, and zero classifications.

We describe this approach in the methods

L515 - 526

Genus dependency on kelp forest habitat

Species maybe observed in a kelp forest but may not strongly or uniquely depend on kelp forest ecosystems for food, shelter, or other benefits. As such, their economic value may not be wholly attributable to a kelp forest ecosystem. We accounted for this fact by creating dependency classes for 187 genera of fish and invertebrates. We then used available information about a genera's habitat preferences and life history to classify genera as either having a high, medium, low, or no dependency on kelp forests. We then corrected for the partial dependency of the medium and low classifications by attributing 2/3rd and 1/3rd of the total economic value respectively to kelp forests. If a species appeared in 5 or fewer surveys, we assigned the genus an economic value of zero as they were likely incidentally observed and not dependent on kelp forests. However, we included all observations in *Lessonia* habitat due to the

limited number of data points available. All relevant data are presented in Appendix 6.

We have also examined the subject again in the discussion, L238-247

The exact contribution of kelp forest habitat to these fisheries services remains an important next step in understanding how kelp forests support food webs⁵⁴ and their related economies. A detailed review paper¹⁵ revealed that kelp forests had a positive effect on fish abundance in 19 of the 24 studies reviewed, a positive effect on crustacean abundance in 4/4 studies, and a positive effect on gastropod abundance in 2/3 studies. Nevertheless, the exact amount of an organism's economic value directly attributable to kelp forests remains unresolved, we partially addressed this issue by assigning genera into high, medium, low, and zero dependency categories and adjusting the economic value based on those classifications (see Methods, Appendix 6). Future work could seek to further address this issue by using more detailed approaches such as stable isotope analysis.

We have also added an appendix with those classifications (Appendix 6).

Finally, we have updated the numbers and figures throughout to reflect the new economic values and steps taken.

We were originally hesitant to “test” the relationship between kelp density and fisheries biomass as the data was not collected for that purpose. Additionally, the subset of data we have for testing this relationship is not representative of the entire dataset. We have therefore removed the regression from the previous Appendix 6 and would prefer to leave dedicated analysis of this question to future work which may be designed to better assess the relationship.

I also noticed somewhat randomly that Appendix 3 column L seems to duplicate into column M at

row 59, shifting the rest of the data set to the right. If this did not happen after all analytical use of these data, such use should probably be checked.

We thank Reviewer 3 for their good eyes. This error is indeed a typo from copying data into the appendix and does not present an issue with the previous analysis. It has now been corrected.

Reviewer #3 (Remarks to the Author):

The revisions are helpful in addressing most of the limitations of the study and incorporating some of the complexities of the underlying economic relationships; good work. Two small things should be added for transparency:

1. the assignment of $1/2/3$, $1/3$, and 0 to the proportion of value generated by the kelp-fishery association appears to be one of expedience (i.e. arbitrary). I understand that exact relationships are not possible, but please note clearly that these are indeed demonstrative rather than reflecting the true expected relationship, so that readers do not assume one or try to tease one out.
2. Similarly, while it is good that there is some effort to align the fisheries values by stage of processing/spoilage risk, it is unclear to me where the 2.5% discounting for processing and spoilage are coming from - this should be clarified, esp. if they have much influence on the overall result (which is also unclear and could be easily built in to the range estimate)

REVIEWER COMMENTS

We thank the reviewer for their continued consideration of our work and provide our individual responses to the reviewer below. We have denoted the sections as follows:

Red: Reviewer's comment

Blue: Our response

Black: In text changes

Reviewer #3 (Remarks to the Author):

The revisions are helpful in addressing most of the limitations of the study and incorporating some of the complexities of the underlying economic relationships; good work. Two small things should be added for transparency:

1. the assignment of 1,2/3, 1/3, and 0 to the proportion of value generated by the kelp-fishery association appears to be one of expedience (i.e. arbitrary). I understand that exact relationships are not possible, but please note clearly that these are indeed demonstrative rather than reflecting the true expected relationship, so that readers do not assume one or try to tease one out.

We have added text describing that these corrections are based on our expert opinion and not on quantitative inputs.

L563

We then used available information about a genera's habitat preferences and life history to *qualitatively* classify genera as either having a high, medium, low, or no dependency on kelp forests. We then corrected for the partial dependency of the medium and low classifications by attributing 2/3rd and 1/3rd of the total economic value respectively to kelp forests. If a species appeared in 5 or fewer surveys, we assigned the genus an economic value of zero as they were likely incidentally

observed and not dependent on kelp forests. However, we included all observations in *Lessonia* habitat due to the limited number of data points available. *These corrections are based on our expert opinion and are subject to change with further analysis (e.g., stable isotope, mixing models, experiments)*. All relevant data are presented in Supplementary Information 6.

2. Similarly, while it is good that there is some effort to align the fisheries values by stage of processing/spoilage risk, it is unclear to me where the 2.5% discounting for processing and spoilage are coming from - this should be clarified, esp. if they have much influence on the overall result (which is also unclear and could be easily built in to the range estimate).

We have further detailed that the corrections were approximated, and the values were made similar to other discounts so that they would not have an outsized influence on the results.

L554

These values are partial corrections and were approximated due to the lack of available information. Given the uncertainty around these values, the discounts were approximated so that they were similar to the other discount values applied (e.g., cost of capital) and thus did not have an outsized influence on the results. Such cost adjustments may be improved upon in future analysis.